

# Primate tooth crown nomenclature revisited

Simon A. Chapple[1] and Matthew M. Skinner[1,2]

[1] School of Anthropology and Conservation, University of Kent at Canterbury, Canterbury, Kent, United Kingdom
[2] Department of Human Evolution, Max Planck Institute for Evolutionary Anthropology, Leipzig, Germany

## ABSTRACT

Cusp patterning on living and extinct primate molar teeth plays a crucial role in species diagnoses, phylogenetic inference, and the reconstruction of the evolutionary history of the primate clade. These studies rely on a system of nomenclature that can accurately identify and distinguish between the various structures of the crown surface. However, studies at the enamel-dentine junction (EDJ) of some primate taxa have demonstrated a greater degree of cusp variation and expression at the crown surface than current systems of nomenclature allow. In this study, we review the current nomenclature and its applicability across all the major primate clades based on investigations of mandibular crown morphology at the enamel-dentine junction revealed through microtomography. From these observations, we reveal numerous new patterns of lower molar accessory cusp expression in primates. We highlight numerous discrepancies between the expected patterns of variation inferred from the current academic literature, and the new patterns of expected variation seen in this study. Based on the current issues associated with the crown nomenclature, and an incomplete understanding of the precise developmental processes associated with each individual crown feature, we introduce these structures within a conservative, non-homologous naming scheme that focuses on simple location-based categorisations. Until there is a better insight into the developmental and phylogenetic origin of these crown features, these categorisations are the most practical way of addressing these structures. Until then, we also suggest the cautious use of accessory cusps for studies of taxonomy and phylogeny.

## INTRODUCTION

Primate dental morphology plays a critical role in reconstructing the phylogenetic relationships (*Wood & Abbott, 1983*; *Bailey, 2000*; *Pilbrow, 2003*; *Skinner, Wood & Hublin, 2009*; *Singleton et al., 2011*), diet (*Kay, 1977*; *Bunn et al., 2011*; *Cooke, 2011*), and ethology of mammalian taxa (*Ungar, 2004*; *Seiffert et al., 2005*). The occlusal surface of tooth crowns in particular often exhibits a complex and variable suite of morphological features that are extensively used in systematics, functional and comparative morphology, and the reconstruction of the evolutionary history of the primate clade. For over a century the study of the occlusal surface of tooth crowns has required a system of nomenclature that

Corresponding author
Simon A. Chapple, sac200@kent.ac.uk

identifies various structures such as cusps and crests. However, over this time the current system of nomenclature has become beset by a number of problems regarding definitions of named structures, multiple names for the same structure, and a growing conflict between current models of tooth crown development (influenced in particular by advances in developmental genetics), and the fundamental paradigm upon which the nomenclature is based. In this study, we review the current nomenclature and its applicability across all the major primate clades, based in particular on our novel investigation of the detailed aspects of mandibular crown morphology at the enamel-dentine junction revealed through microtomography.

The most widely used and established system of nomenclature was initially developed from Edward Drinker Cope's work on the evolution of mammalian tooth form, and Henry Fairfield Osborn's elaboration of these ideas into a nomenclature (*Cope, 1883*; *Osborn, 1888*). Cope's work described a model for the evolution and development of tribosphenic, multicuspid molars from the primitive cone-shaped teeth of mammalian ancestors. According to the model, the ancestral condition was haplodonty; a single, cone-shaped structure that *Osborn (1888)* and *Osborn (1907)* called the protocone for the upper dentition, and protoconid for the lower dentition. Two additional cusps then developed from this cone, initially in mesial and distal orientation to the protocone(id), and were named the paracone(-id) and metacone(-id), respectively. From this triconodont configuration, Cope believed the paracone and metacone of the upper teeth migrated in a buccal direction, while the protocone moved lingually, creating a V-shaped symmetrodont configuration. In the lower molars, a similar migration of cusps was thought to have occurred. However, in this case the paraconid and metaconid rotated lingually relative to the protoconid, creating a reversed triangle configuration between upper and lower dentitions. In the quadritubercular upper molar, a fourth cusp distal to the protocone later formed on a low shelf and was named the hypocone. In the lower molars, a low shelf also formed distal to the symmetrodont triangle, from which developed the entoconid on the lingual margin, the hypoconid on the buccal margin, and the hypoconulid on the distal margin. In addition to the primary cusps of the mammalian molar, secondary features of upper and lower molars were named using the prefixes associated with their neighbouring primary cusps, along with an appropriate suffix to denote the type of feature in question (conules or conulids for cusps, and crista or cristid for crests). For crests, these names are further preceded with a pre- or post-connotation to indicate the location or 'direction' of the crest relative to the anterior-posterior orientation of the tooth and associated cusp.

Unfortunately, as researchers began to identify and study new fossil species, it became clear that some stages of evolution described by Cope and Osborn's model did not form a strict phylogenetic sequence as they had assumed. Additionally, it is now known that multicuspid configurations developed independently in several therapsid groups, and that the cheek teeth of the earliest known mammals were not haplodont (*Patterson, 1956*; *Butler, 1978*). Ultimately, Cope and Osborn had misinterpreted cusp homologies and got the order of cusp appearance wrong. As a consequence, Osborn's nomenclature, which was originally intended to denote evolutionary processes and historical homology, was found to be flawed. Palaeontological evidence now indicates that the primitive cusp of early upper and lower

molars is the mesio-buccal cone, or the paracone and protoconid of Osborn's nomenclature (*Butler, 1978*). Furthermore, the mesial cusp of the triconodont configuration is not the paracone seen in extant taxa, but is now recognised as the stylocone (*Butler, 1978*). While the metacone is still homologous with the metaconid, the other upper and lower primary cusps of the same prefix are no longer considered homologous. Such fundamental flaws in nomenclature resulted in what *Hershkovitz* (*1971*, p. 95) described as the "corruption of dental evolutionary thought through use of similar terms for non-homologous upper and lower dental elements, and dissimilar terms for the homologous element(s)".

Since the introduction of Osborn's nomenclature, numerous alternative systems and names have been devised and adopted, either to address some of the known issues of homology that had been recognised in Osborn's terminology, due to a perceived better representation and corresponding description of the feature in question, or in an attempt to communicate a structure in a way that is free of developmental implication (*Remane, 1960*; *Van Valen, 1966*). In some cases, this involved a substantial revision and a proposal of a new system (*Vandebroek, 1961*), while in other cases it simply involved the introduction of new terms as they were recognised and studied (*Dahlberg, 1950*). In 1961, Vandebroek proposed a new system of nomenclature for tribosphenic molars that attempted to address some of the issues in Osborn's terminology (*e.g.*, suggesting the term 'eocone' for the 'paracone', and 'epicone' for the 'protocone'). However, despite some support and use of this system in the academic literature, it was not widely accepted. A decade later, *Hershkovitz (1971)* proposed his revision of the nomenclature that suggested to serve as a "master plan of coronal pattern of upper and lower euthemorphic molars" (p. 135). This system maintained some of Osborn's terms, adopted the eocone and epicone of *Vandebroek (1961)*, and introduced several new terms and previously unstudied dental elements. This resulted in a nomenclature that acknowledged 92 different features on the upper and lower molar crown. While several aspects of this amalgamated nomenclature were adopted, many previously proposed terms were preferred and maintained, resulting in a mosaic, interchangeable, and highly inconsistent nomenclature that varies in its use of the many positional, Osbornian, numerical, Latin, and clinical terms that currently exist (Fig. 1).

Due to the convoluted history of the current nomenclature, the above noted proliferation, mixing, and multiplication of terms, and the influence that individual researchers working on particular mammalian groups has had on the current nomenclature, a number of issues should be acknowledged and addressed. First, early systems only described the basic morphologies, did so with simple descriptions or diagrams, and the original description is often difficult to reconcile with more recent systems. Second, many nomenclatures attempt to apply their systems to taxonomically broad groups (*e.g.*, all primates/mammals). However, while these, in principal, can allow for discourse on the evolution and homology of dental crown features across wide groups, in reality they become burdened by inconsistencies or inapplicability to the variation that is present in the groups they are applied to. For example, terminologies initially created based on observations of crown morphology of a specific clade of mammal (*e.g.*, *Gregory, 1916*), may be unsuitable for all primates. Third, many of the current systems do not provide enough topographical information (both directional and positional) to ensure their accurate use

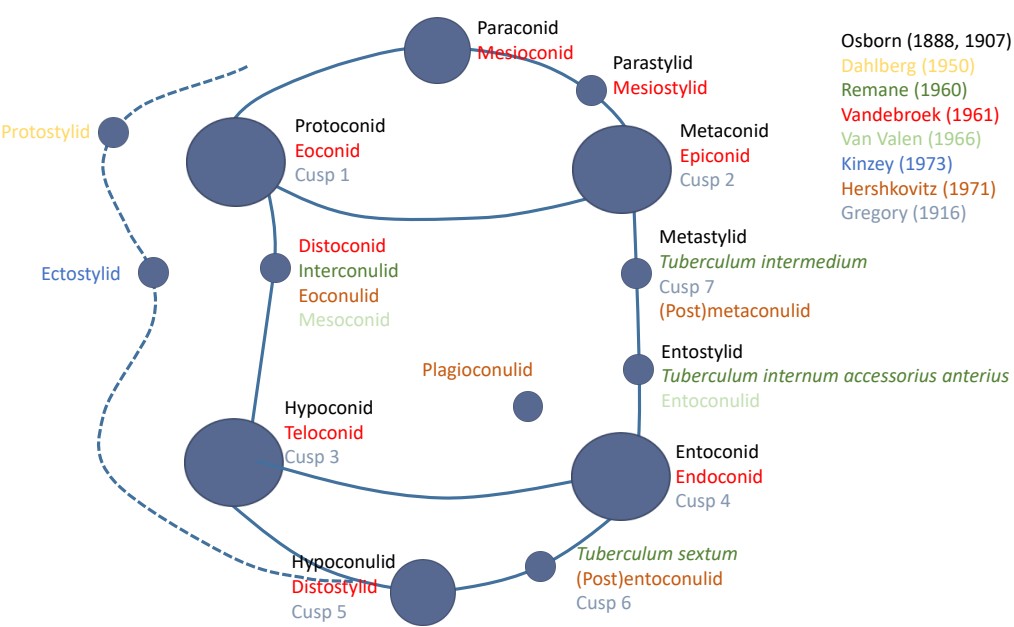

**Figure 1** **Schematic diagram summarizing some of the commonly used nomenclature terms used to identify primary and secondary cusps on the molar crown.** Colours of each term correspond to an author that either originally introduced the term, or has since championed the use of the term in studies of dental morphology.

(*e.g.*, the interconulid and the varied use of this term for different morphological structures). Similarly, some systems are very complex and require the identification of a single and specific term from a diagram with many closely-positioned but distinct features (*e.g.*, Hershkovitz's many ectostylid forms). Additionally, some terms reflect assumptions about the developmental origin of the feature and/or their association with adjacent features, when we lack direct evidence of an actual developmental link. Fourth, some systems still maintain names for features that are associated with an extinct nomenclature, such as the inappropriate use of the term eoconulid if one is not using the term eoconid for the mesio-buccal primary cusp (*e.g.*, *Vandebroek, 1961*). Finally, as new terms were often introduced as direct equivalents to previously named features, authors have attempted to provide lists of current terms and synonyms that are considered equivalent. However, due to many of the factors discussed above, these synonyms are not always accurate and therefore introduce further error into the system (*Swindler, 2002* and the many synonymic errors in Table 1.2).

Of particular relevance to the modern application of tooth crown nomenclatures is our growing understanding on the developmental processes that control cusp patterning on tooth crowns. In particular, advances in our understanding of multicupid tooth development suggest that accessory cusp presence and expression may not be predetermined, and may instead be based on a number of upstream factors such as the size, shape and location of the surrounding primary cusps and tooth germ (*Jernvall, 2000*; *Jernvall & Thesleff, 2000*). This process has been called the patterning cascade model of

cusp development (*Polly, 1998*), and suggests that accessory cusp initiation and patterning may be an iterative process that is particularly sensitive to small variations in the shape and positionings of earlier-forming, neighbouring cusps. This iterative nature of cusp pattering is important when evaluating the appropriateness of detailed aspects of a nomenclature. For example, most systems of nomenclature allow for the presence of between four and seven cusps on hominoid lower molars. However, *Skinner et al. (2008)* showed that up to nine cusps can be present, with many of these displaying variable degrees of expression and positioning along the occlusal surface. Traditionally, practitioners have used terms such as 'double' to denote the presence of two cusps in a particular region, however, these terms do not appear to accurately reflect their developmental origin (*Skinner et al., 2008*).

A major advance in our ability to visualize and interpret tooth crown morphology for the purpose of revisiting primate crown nomenclature has come from high resolution imaging of the enamel-dentine junction (EDJ). The EDJ preserves the morphology of the basement membrane of the developing tooth germ prior to mineralization (*Nager, 1960*; *Kraus & Jordan, 1965*), and therefore represents the first stage of crown development in which cusps and crests appear. Imaging the EDJ has been shown to not only to record the presence and size of dental crown features with increased accuracy, but also allows greater insights into the developmental origin and individual morphology of dental traits (*Skinner et al., 2008*; *Skinner et al., 2010*). For example, *De Pinillos et al. (2014)* and *Martin et al. (2017)* addressed the potential taxonomic and phylogenetic utility of dental non-metric traits at the EDJ in Late Pleistocene hominins and modern humans, while others have addressed concerns regarding the homology and identity of certain crown features previously described at the outer enamel surface in fossil hominins (*Anemone, Skinner & Dirks, 2012*; *Ortiz et al., 2017*; *Davies et al., 2021*).

The aim of this study is to critically evaluate the current nomenclature scheme used for primate molar crowns, using micro-computed tomography (micro-CT) to image the EDJ in a taxonomically broad sample of primate mandibular first and second molars. The study has three objectives: First, to document variation in cusp patterning within major clades of the order Primates; second, to assess the applicability of the current nomenclature to each clade and to propose clade specific nomenclatures when appropriate; third, to present an updated approach to the use of nomenclature schemes for the purpose of primate systematics.

## MATERIALS & METHODS

### Materials

The study sample is shown in Table 1. The sample consists of mandibular first and second molars from 420 specimens, representing 71 primate species (a complete list of specimens is provided in the Supporting Information). The study sample was selected to include species from all major clades. Sex was not considered as there is no evidence that it impacts the presence of cusps. Sample sizes for some species are low due to difficulties in identifying and microCT scanning specimens with unworn mandibular molars (this is particularly challenging as most primate species are relatively thin-enamelled). Specimens with low

**Table 1  Primate species used in this study divided into clades relevant to their tooth crown morphology.**

| Strepsirrhini | n | Ceboidea | n | Cercopithecidae | n | Hominoidea | n |
|---|---|---|---|---|---|---|---|
| **Lemuridae** | | **Callitrichinae** | | **Cercopithecini** | | **Hominidae** | |
| *Prolemur simus* | 1 | *Cebuella pygmaea* | 1 | *Erythrocebus patas* | 2 | *Pan troglodytes* | 55 |
| *Hapelemur griseus* | 2 | *Callithrix jacchus* | 3 | *Chlorocebus aethiops* | 4 | *Pan paniscus* | 22 |
| *Eulemur fulvus* | 2 | *Leontopithecus rosalia* | 2 | *Miopithecus talapoin* | 2 | *Homo sapiens* | 56 |
| *Varecia variegata* | 2 | *Leontopithecus chrysopygus* | 2 | *Cercopithecus mitis* | 6 | *Gorilla gorilla* | 12 |
| | | *Saguinus mystax* | 2 | | | *Pongo pygmaeus* | 40 |
| **Lepilemuridae** | | *Saguinus oedipus* | 2 | **Papionini** | | | |
| *Lepilemur leucopus* | 2 | | | *Macaca fascicularis* | 24 | **Hylobatidae** | |
| *Lepilemur mustelinus* | 1 | **Aotinae** | | *Macaca fuscata* | 11 | *Hylobates muelleri* | 4 |
| | | *Aotus sp.* | 10 | *Macaca arctoides* | 2 | *Hylobates agilis* | 2 |
| **Cheirogaleidae** | | | | *Macaca sylvanus* | 2 | *Hylobates lar* | 1 |
| *Phaner furcifer* | 2 | **Cebinae** | | *Lophocebus albigena* | 5 | *Hoolock sp.* | 1 |
| *Microcebus sp.* | 2 | *Saimiri sp.* | 2 | *Papio anubis* | 15 | | |
| *Cheirogaleus sp.* | 3 | *Cebus olivaceus* | 1 | *Theropithecus gelada* | 2 | | |
| | | *Cebus albifrons* | 4 | *Mandrillus sphinx* | 3 | | |
| **Indriidae** | | *Cebus capucinus* | 1 | *Mandrillus leucophaeus* | 1 | | |
| *Propithecus diadema* | 1 | *Sapajus apella* | 5 | *Cercocebus torquatus* | 2 | | |
| *Indri indri* | 2 | | | | | | |
| *Avahi laniger* | 2 | **Pitheciinae** | | **Colobinae** | | | |
| | | *Cacajao calvus* | 3 | *Nasalis larvatus* | 4 | | |
| **Galagidae** | | *C. melanocephalus* | 5 | *Semnopithecus entellus* | 2 | | |
| *Galago senegalensis* | 3 | *Chiropotes satanas* | 9 | *Trachypithecus cristatus* | 5 | | |
| *Otolemur garnettii* | 2 | *Pithecia pithecia* | 6 | *Trachypithecus vetulus* | 2 | | |
| *Euoticus elegantulus* | 2 | | | *Presbytis comata* | 1 | | |
| | | **Callicebinae** | | *Presbytis melalophos* | 4 | | |
| **Lorisidae** | | *Callicebus moloch* | 4 | *Piliocolobus pennantii* | 3 | | |
| *Loris tardigradus* | 2 | | | *Colobus guereza* | 10 | | |
| *Nycticebus coucang* | 3 | **Atelinae** | | | | **Tarsioidea** | **n** |
| *Perodicticus potto* | 2 | *Ateles geoffroyi* | 9 | | | *Tarsius spectrum* | 2 |
| *Arctocebus calabarensis* | 2 | *Alouatta seniculus* | 8 | | | *Tarsius syrichta* | 1 |

to moderate attrition were included as it did not impact our ability to identify particular crown features with confidence. Due to these small sample sizes the frequency of traits is not assessed.

## Methods

Specimens were scanned on a number of different microCT systems including beamline ID 19 at the European Synchrotron Radiation Facility (ESRF, Grenoble, France), or a BIR ACTIS 225/300 or SkyScan 1,172 microtomographic scanner at the Department of Human Evolution, Max Planck Institute for Evolutionary Anthropology. Scanning was conducted under standard operating conditions following established protocols (*Olejniczak et al., 2007*). Scan resolution varied between 10–60 microns. This resolution is sufficient to identify small crown features, although it is acknowledged that very tiny dentine horns

may be poorly resolved making their assessment difficult. Individual molars were initially cropped in Avizo 9.0 (http://www.thermofisher.com). To facilitate tissue segmentation, image stacks of each molar were then filtered using a 3D mean-of-least-variance filter with a kernel size of one. This process sharpens the boundaries between enamel and dentine (*Schulze & Pearce, 1994*), allowing for a clearer segmentation of tissue types. Filtering was implemented using MIA open-source software (*Wollny et al., 2013*). Enamel and dentine were segmented in Avizo 6.3 using a seed growing watershed algorithm employed *via* a custom Avizo plugin, before being manually checked. After segmentation, triangle-based surface models of the EDJ were produced using the generate surface module in Avizo, and then saved in polygon file format (.ply).

For the purpose of evaluating current nomenclatures and creating new nomenclatures, species are grouped at the highest taxonomic level possible where similarities in mandibular crown morphology allow. In the results section for each group, we first create a 'current variation' schematic based on a review of the published literature that acknowledges crown features that have previously been observed and discussed. In many cases, this has never been assessed at the taxonomic levels that we identify here as relevant and necessary. We then report on crown feature variation at the EDJ present in the study sample and propose a 'new' expected variation schematic for each group. *Davies et al. (2021)* recently proposed the adoption of conservative terms for accessory cusps on hominin lower molars due to difficulties in defining variations in dentine horn presence on the lingual and distal marginal ridge of the EDJ. Specifically, they adopted the terms distal accessory cusp(s) and lingual accessory cusp(s) instead of cusp 6 and cusp 7, respectively. Our observations of accessory cusp presence on lower molars in this study sample reinforce the utility of the use of such generic terms and here we expand this to both the mesial and buccal marginal ridges of the EDJ, as well as, the cingulum (Fig. 2). As a result, we propose the following terms to classify accessory cusps (AC) on the marginal ridge of the EDJ that runs between the four primary cusps (protoconid, metaconid, entoconid, hypoconid): DM—distal margin; LM—lingual margin; MM—mesial margin; BM—buccal margin. Additionally, we propose the following terms to classify accessory cusps on the cingulum: BC—buccal cingulum; LC—lingual cingulum.

## RESULTS

### Strepsirrhini

In the family Lemuridae, the two mesial primary cusps are compressed bucco-lingually, are set close together, and are connected by a transverse crest. A distinct crest also traverses down the mesial slope of the protoconid, where it eventually turns and proceeds disto-lingually as a broad ledge to the base of the metaconid. The hypoconid and entoconid are not connected by a crest, and the talonid basin is shallow and circumscribed by the marginal ridge. The entoconid is absent in *Varecia variegate* (*Schwartz & Tattersall, 1985*). While a paraconid was present in fossil notharctids of the early Eocene (*Gregory, 1920*), it is absent in extant Lemuridae. *Cuozzo & Yamashita* state that lemurids "have lost the paraconid and lack a hypoconulid" (*2006*, p. 76); however *Swindler* (*2002*, p. 69) describes the presence

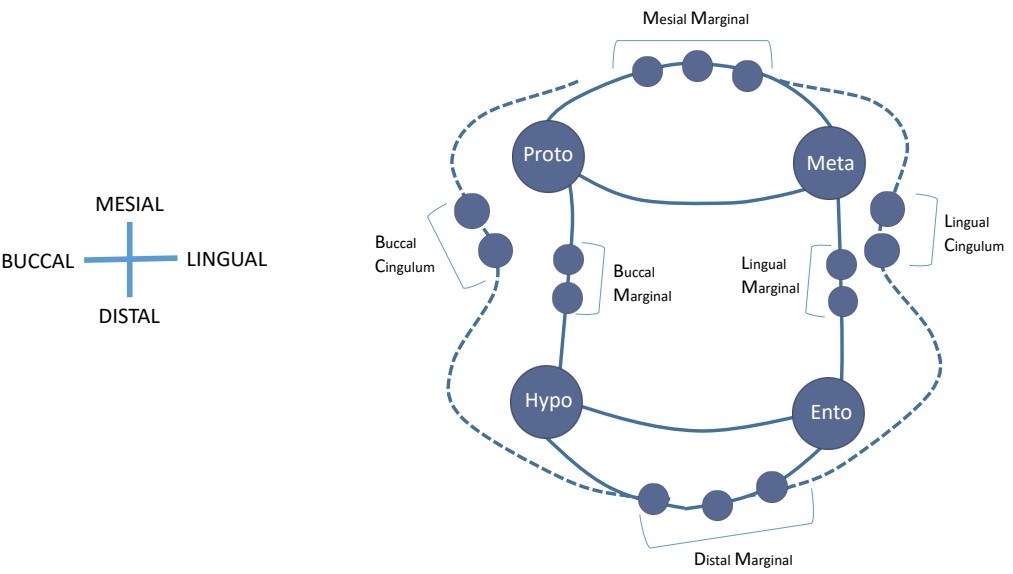

**Figure 2** **Schematic diagram of the proposed terms used to classify accessory cusps (AC) on the marginal ridge of the EDJ.** DMAC, distal margin; LMAC, lingual margin; MMAC, mesial margin; BMAC, buccal margin. Additionally, we propose the following terms to classify accessory cusps on the cingulum: BCAC, buccal cingulum; LCAC, lingual cingulum.

of a "distal heel-like process" in some members of this clade and considers it to be a true hypoconulid. *Swindler (2002)* also reports the presence of a *tuberculum intermedium* in all six of his *Hapalemur griseus* lower first molars, and in five of the six second molars. Similarly, *Schwartz & Tattersall (1985)* describe a thickening of the postmetacristid in *Lemur catta* molars that results in the expression of what they term a 'metastylid'. Some form of buccal cingulid expression is noted in all Lemuridae molars. Thus, the current variation scheme can be summarized with four primary cusps, a hypoconulid on the distal ridge, and a *tuberculum intermedium* on the lingual ridge (Fig. 3A). Our observations support descriptions of a cusp on the distal marginal ridge of some specimens (Fig. 3D), which we identify and label as a DMAC in the new schematic for Lemuridae (Fig. 3B). Additionally, we also corroborate the reports of accessory cusp presence on the lingual marginal ridge, with LMACs observed in the *Prolemur simus* and *Eulemur fulvus* specimens in our sample (Figs. 3E and 3F respectively). The image of the *Varecia variegate* specimen (Fig. 3G) demonstrates the absence of an entoconid in this taxon, but the expression of several LMACs along the marginal ridge.

The family Lepilemuridae consists of only one extant genus, *Lepilemur*. The molars have four primary cusps, with a diagonal transverse crest connecting a mesially-placed protoconid to a comparatively more distally-placed metaconid. *Swindler (2002)* reports that the crest travelling down the mesial slope of the protoconid may connect with the mesio-buccal elevation of the cingulid, forming a mesiostylid where to two crests join. Descriptions also note a distinct and complete buccal cingulid, and a strong cristid obliqua that terminates on the lingual side of the protoconid (*Schwartz & Tattersall, 1985*). While

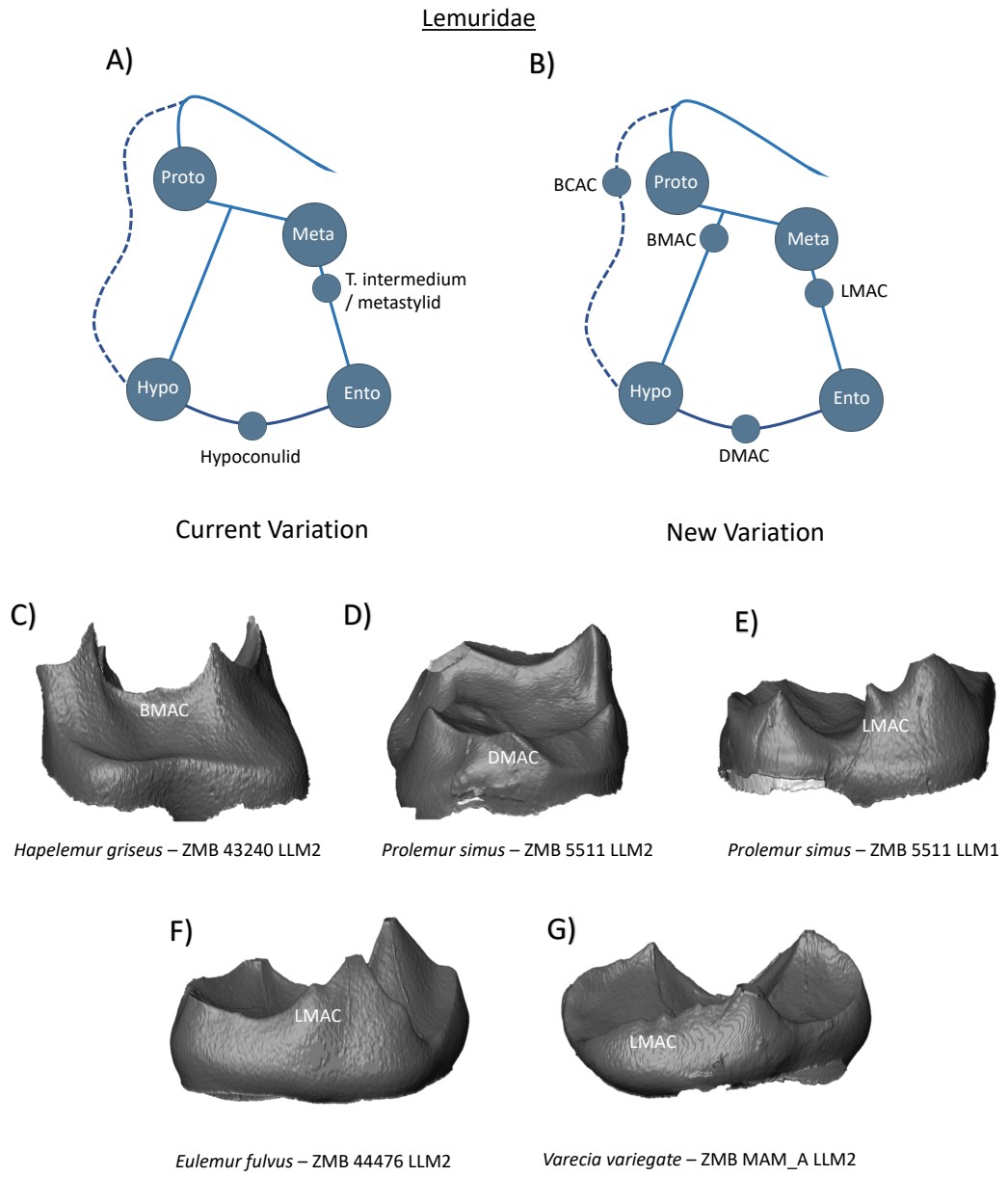

Figure 3 **Crown patterning in Lemuridae.** (A) Current variation schematic for Lemuridae, based on a review of the published literature. (B) New variation schematic for Lemuridae, based on observations at the enamel-dentine junction. (C) *Hapelemur griseus* lower second molar with BMAC expression. (D) *Prolemur simus* lower second molar with DMAC expression. (E) *Prolemur simus* lower first molar with LMAC expression. (F) *Eulemur fulvus* lower second molar with LMAC expression. (G) *Varecia variegate* lower second molar with no discernible entoconid, but several LMAC cusps.

there has been limited commentary regarding accessory cusps expression in this clade, the potential presence and identification of a fifth cusp has been extensively discussed (*James, 1960*; *Seligsohn & Szalay, 1978*; *Schwartz & Tattersall, 1985*). Unfortunately, these discussions remain focused on hypoconulid expression in lower third molars, and there

is little mention of similar manifestations in first or second molars. A further topic of debate is the identification of the cusp positioned distal to the metaconid. Based on comparisons with other strepsirhines like indriids and *Copelemur*, *Schwartz & Tattersall (1985)* suggest that the entoconid is absent in *Lepilemur* and that the cusp distal to the metaconid is a metastylid. *Swindler* (*2002*, p. 72) argues that "an obvious developmental groove" that separates the two cusps at the outer enamel surface would suggest that *Schwartz & Tattersall*'s (*1985*) metastylid is the entoconid. The current variation schematic for this clade therefore includes a hypoconulid, and a potential entoconid or metastylid (Fig. 4A). Our two *Lepilemur* specimens do not demonstrate the presence of a mesiostylid or hypoconulid (Figs. 4C–4H). Nevertheless, due to sample size limitations, we cannot rule out their existence in other individuals and therefore include MMAC and DMAC features in the new schematic for Lepilemuridae (Fig. 4B). Figures 4D and 4G reveal lingual crown morphology at the EDJ surface, and in particular, the positioning and appearance of the cusp distal to the metaconid. Unlike LMAC expression in our Lemuridae sample, which are positioned on the distal slope of the metaconid, the disto-lingual cusp in *Lepilemur* sits further back on the crown and appears developmentally distinct from the metaconid. As such, we label it the entoconid in our new schematic (Fig. 4B). No other features of relevance were identified.

In the family Cheirogaleidae, the protoconid and metaconid are closely positioned, and are connected by a transverse crest that separates a small trigonid from a spacious talonid basin. In *Cheirogaleus major*, *Schwartz & Tattersall* (*1985*, p. 38) remark that the molar cusps "lack virtually all detail and delineation of individual features", and that structures can therefore only be discussed in a vague sense. Descriptions of hypoconulid expression in cheirogalids are restricted to *Microcebus* lower third molars (*James, 1960*; *Schwartz & Tattersall, 1985*; *Cuozzo et al., 2013*), while *Swindler (2002)* reports the presence of a buccal cingulid in all lower molars. Thus, the current variation scheme can be summarized as a relatively simple tooth crown with only four primary cusps (Fig. 5A). Our assessment of *Cheirogaleus* molars is in agreement with the comments regarding a lack of discernible features by *Schwartz & Tattersall (1985)*. Even from the detail provided by high resolution images of the EDJ, *Cheirogaleus* molars lack any clearly defined cusps (Figs. 5C and 5D). As such, commentary regarding potential accessory cusp expression in this taxon is avoided. Contrastingly, in our *Phaner furcifer* specimen, a prominent buccal cingulid accompanies four well-defined primary cusps. Figure 5E demonstrates the presence of a large BCAC on the buccal cingulid, which represents the only addition to the new expected variation schematic for Cheirogaleidae molars (Fig. 5B).

The molars of Indriidae have four well-formed primary cusps. In the lower first molars, the protoconid is mesial to the metaconid, and the mesial marginal ridge is incomplete mesio-lingually. At the terminus of this incomplete ridge, *Swindler (2002)* mentions the presence of a parastylid. In the lower second molars, the protoconid and metaconid are positioned at similar mesio-distal positions on the crown, the mesial marginal ridge is complete, and the cusps are also connected by a faint transverse crest. Hypoconulid presence is only referenced in relation to lower third molars (*Swindler, 2002*). *Bennejeant (1936)* mentions the frequent appearance of a *tuberculum intermedium* on the distal surface

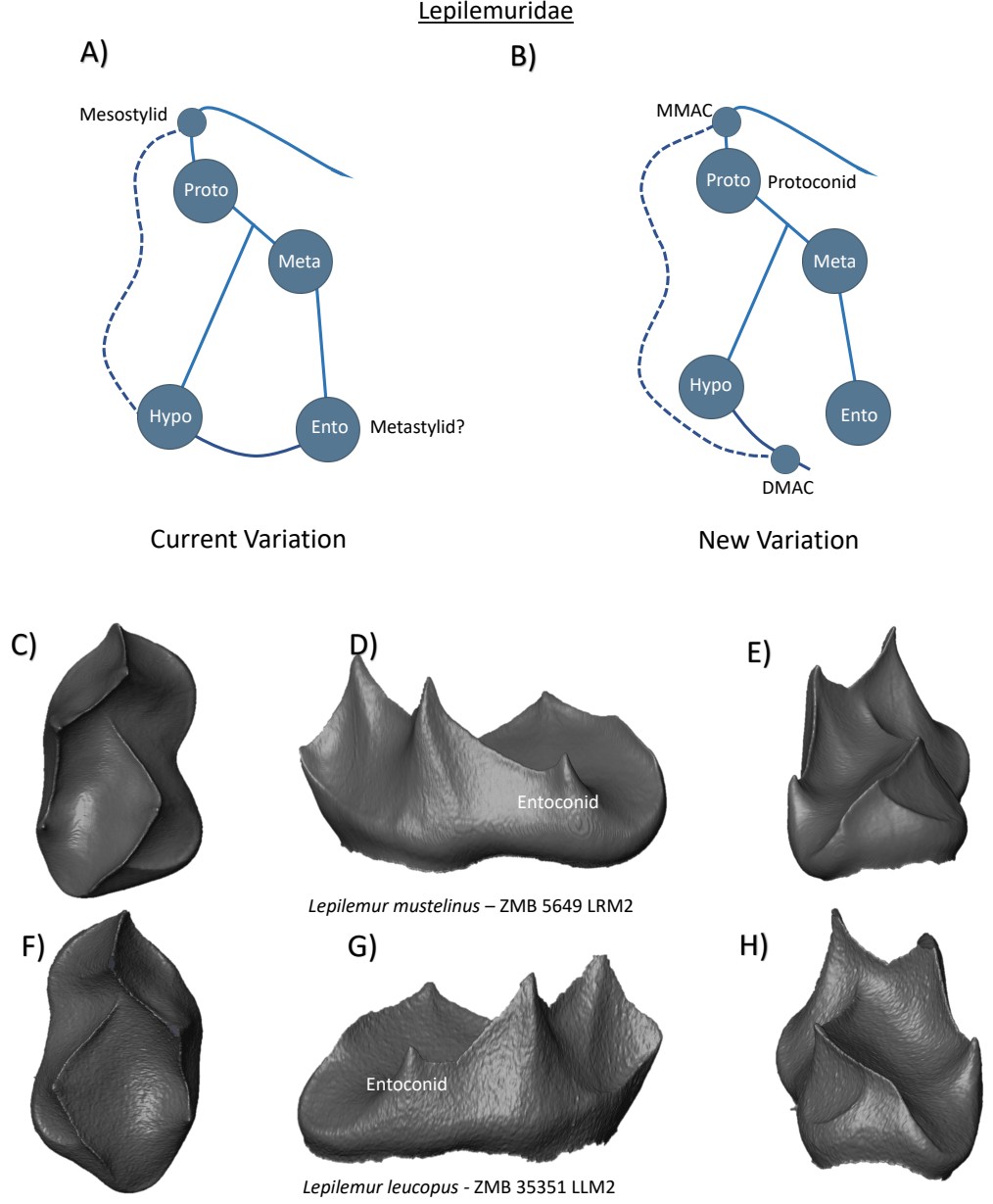

**Figure 4 Crown patterning in Lepilemuridae.** (A) Current variation schematic for Lepilemuridae, based on a review of the published literature. (B) New variation schematic for Lepilemuridae, based on observations at the enamel-dentine junction. (C) *Lepilemur mustelinus* lower second molar with no discernible features of interest. (D) *Lepilemur mustelinus* lower second molar with a short entoconid cusp. (E) *Lepilemur mustelinus* lower second molar with no discernible features of interest. (F) *Lepilemur leucopus* lower second molar with no discernible features of interest. (G) *Lepilemur leucopus* lower second molar with a short entoconid cusp. (H) *Lepilemur leucopus* lower second molar with no discernible features of interest.

of the metaconid in *Avahi*, while *Schwartz & Tattersall (1985)* also report the presence of an equivalent feature at the terminus of a thick postmetacristid. The current variation scheme provided therefore incorporates both parastylids and *tuberculum intermediums*, in addition

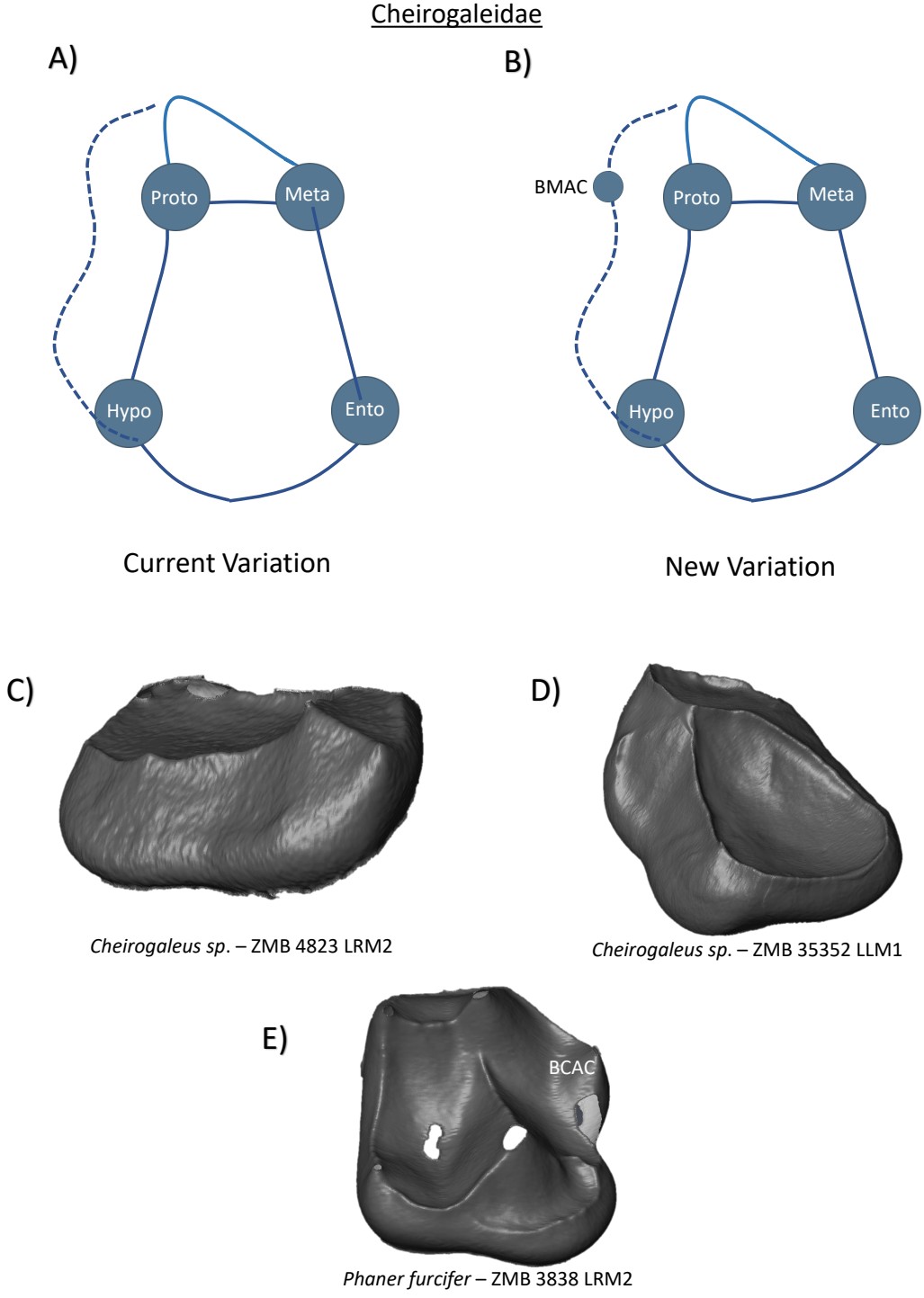

Cheirogaleidae

A)

Current Variation

B)

New Variation

C)

*Cheirogaleus sp.* – ZMB 4823 LRM2

D)

*Cheirogaleus sp.* – ZMB 35352 LLM1

E)

*Phaner furcifer* – ZMB 3838 LRM2

**Figure 5** **Crown patterning in Cheirogaleidae.** (A) Current variation schematic for Cheirogaleidae, based on a review of the published literature. (B) New variation schematic for Cheirogaleidae, based on observations at the enamel-dentine junction. (C) *Cheirogaleus sp.* lower second molar with no discernible features of interest. (D) *Cheirogaleus sp.* lower first molar with no discernible features of interest. (E) *Phaner furcifer* lower second molar with BCAC expression.

to four primary cusps (Fig. 6A). It should be noted that both in the relative positioning of the cusps, and the arrangement of the marginal ridge, these schematic diagrams remain relatively accurate for lower first molars but do not reflect the general shape and patterning of some Indriidae lower second molars. Our observations of Indriidae molar morphology at the EDJ found no evidence for hypoconulid (or DMAC) expression in our sample. Figure 6C demonstrates the incomplete mesial marginal ridge in an *Indri indri* first molar, while Fig. 6D shows the presence of a small MMAC on the complete marginal ridge of a second molar. No LMACs were observed in this sample, however Fig. 6E demonstrates the thick lingually deflected postmetacristid in *Avahi* described by *Schwartz & Tattersall (1985)*. Despite not observing a lingual accessory cusp in this sample, the new schematic for Indriidae includes the presence of an LMAC (Fig. 6B), along with the MMAC observed in the *Indri* lower left second molar.

In the family Galagidae, the first and second molars have four primary cusps, and a well-developed cristid obliqua (*Swindler, 2002*). As is common within many strepsirrhini clades, a compressed crest descends down the mesial face of the protoconid before angling back toward the metaconid as a broad, horizontal ledge. Discussion regarding hypoconulid presence is limited to lower third molars, and mention of a centrally emplaced heel in *Galago senegalensis,* and more lingually displaced heel in *Galago alleni* (*Schwartz & Tattersall, 1985*). *Schwartz & Tattersall (1985)* also describe the presence of a protostylid in *Euoticus elegantulus*, as well as a low conulid on the cristid obliqua at the base of the protoconid in *Galago crassicaudatus*. The current variation schematic therefore depicts four primary cusps, a protostylid, and an unnamed conulid on the buccal margin (Fig. 7A). From our observations, we demonstrate the presence of multiple DMACs in a *Galago senegalensis* first molar (Fig. 7C). In the same specimen we also identify an MMAC positioned where the mesial marginal ridge sharply bends towards the metaconid (Fig. 7D). We find no evidence of a protostylid in any Galagidae specimen, however the lingual positioning of the cristid obliqua and buccal flaring of the protoconid would appear to be consistent with morphological conditions commonly associated with protostylid presence. As such, a BCAC is included in the new schematic (Fig. 7B). In relation to the conulid observed by *Schwartz & Tattersall (1985)* on the cristid obliqua of *Galago crassicaudatus*, we demonstrate the similar presence of a dentine horn distal to the protoconid in *Euoticus elegantulus* that we more appropriately label as a BMAC (Fig. 7E).

The family Lorisidae are stated to have four well-developed cusps, a hypoconulid that is restricted to third molars, a prominent cristid obliqua, and a transverse crest that separates the trigonid basin from a spacious talonid basin (*Swindler, 2002*). In *Arctocebus calabarensis*, *Swindler (2002)* reports a paracristid that extends down the protoconid and ends as a mesiostylid (Fig. 8A). While the presence of a cusp at the terminus or turning point of the paracristid is common in strepsirrhini, we find no evidence of a dentine horn in our current Lorisidae sample. Based on previous observations however, we include the presence of a mesio-bucally positioned MMAC in the new schematic (Fig. 8B). Of particular note and relevance in this clade are the observations of distal and disto-lingual accessory cusp expression. *Loris tardigradus, Arctocebus calabarensis, Nycticebus coucong* and *Perodicticus potto* specimens all demonstrate single and/or multiple DMAC expression

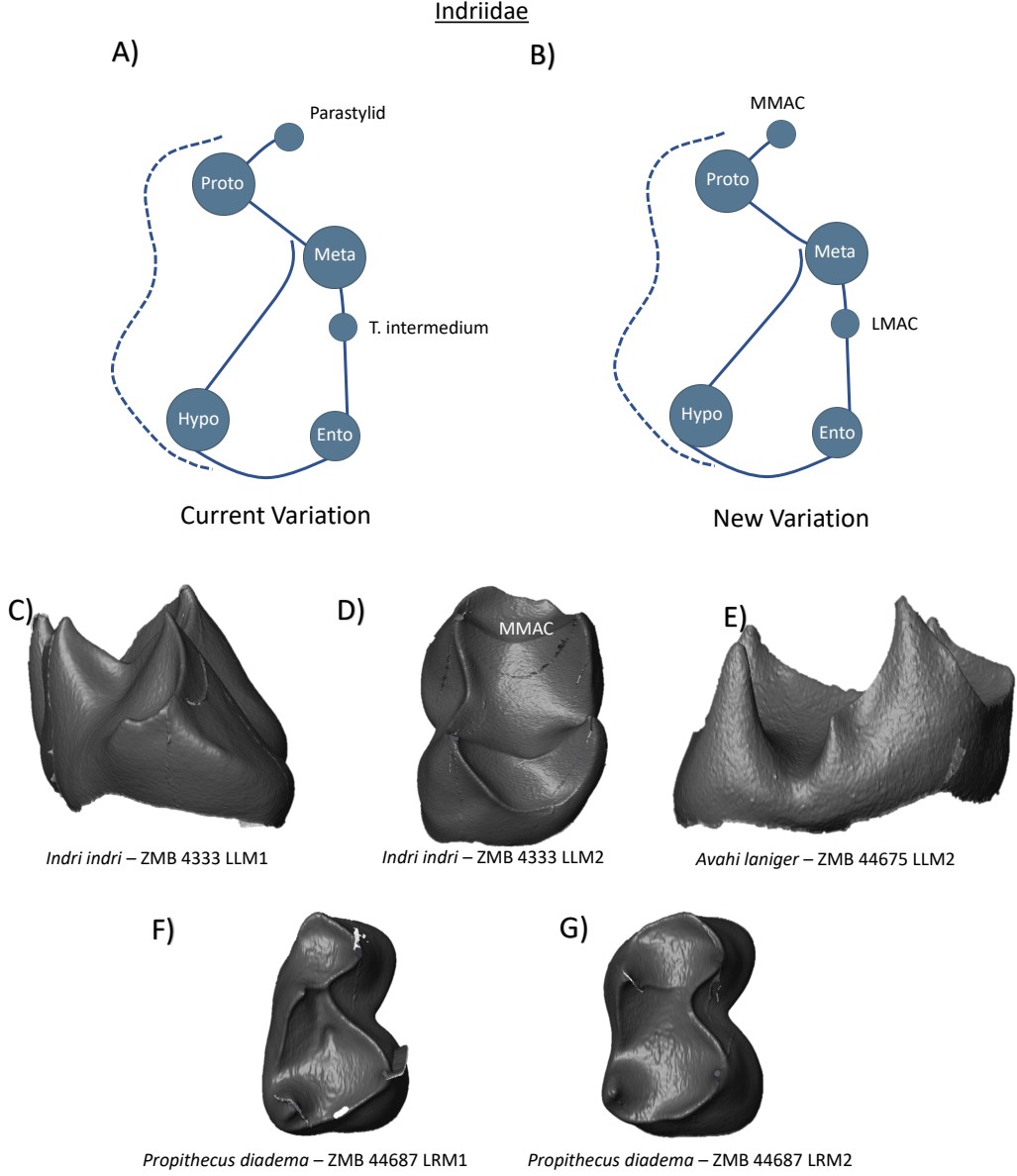

**Figure 6  Crown patterning in Indriidae.** (A) Current variation schematic for Indriidae, based on a review of the published literature. (B) New variation schematic for Indriidae, based on observations at the enamel-dentine junction. (C) *Indri indri* lower first molar with no discernible features of interest . (D) *Indri indri* lower second molar with MMAC expression. (E) *Avahi laniger* lower second molar with no discernible features of interest. (F) *Propithecus diadema* lower first molar with no discernible features of interest. (G) *Propithecus diadema* lower second molar with no discernible features of interest.

on the distal border of the crown (Figs. 8C–8G). In one of the *Perodicticus potto* specimens (Fig. 8F), it may be unclear which of the two disto-lingual cusps is the entoconid, and therefore it is unclear whether this molar has a double DMAC configuration, or a single DMAC and LMAC pattern. However, as the entoconid is situated in a strongly lingual

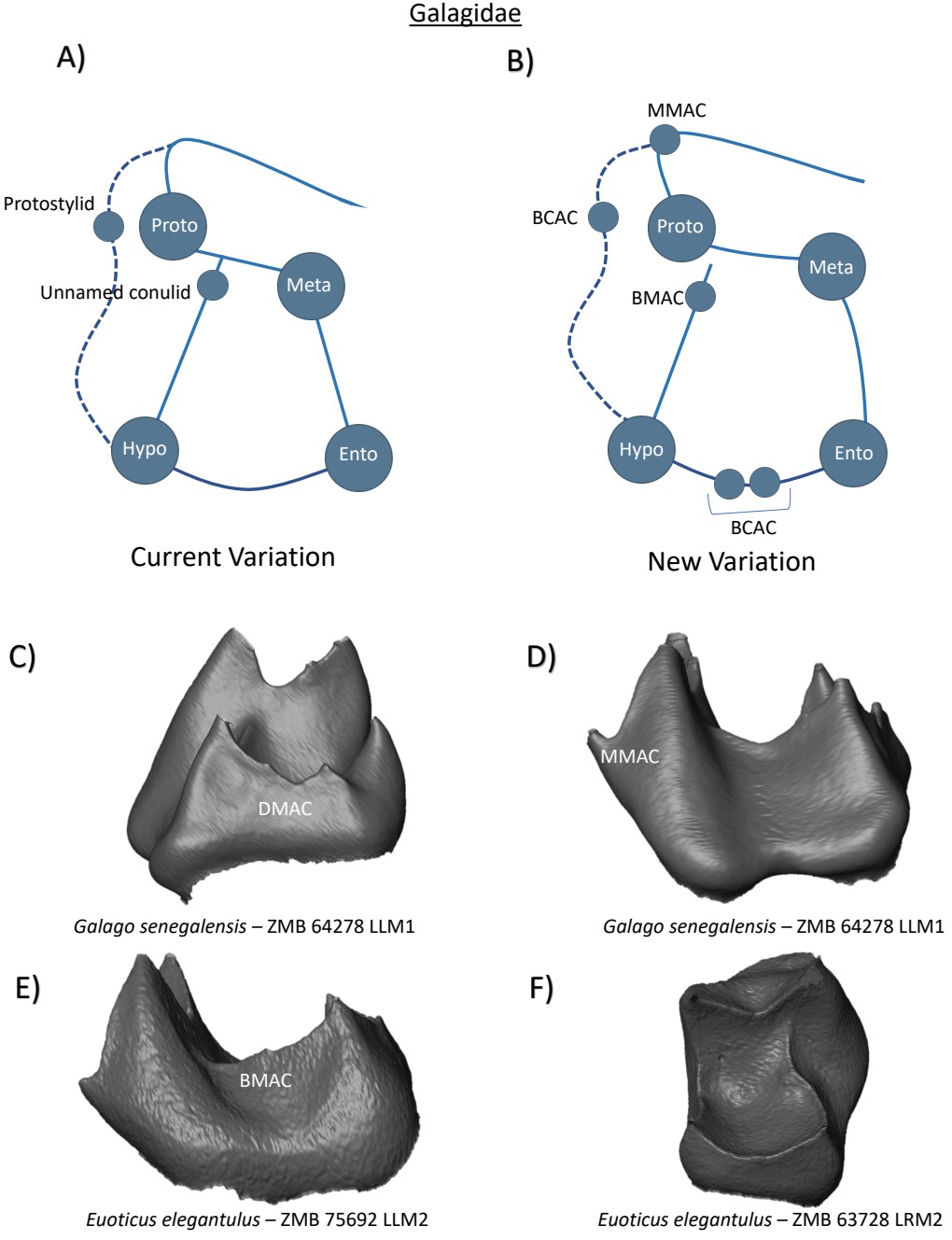

**Figure 7 Crown patterning in Galagidae.** (A) Current variation schematic for Galagidae, based on a review of the published literature. (B) New variation schematic for Galagidae, based on observations at the enamel-dentine junction. (C) *Galago senegalensis* lower first molar with DMAC expression. (D) *Galago senegalensis* lower first molar with MMAC expression. (E) *Euoticus elegantulus* lower second molar with BMAC expression. (F) *Euoticus elegantulus* lower second molar with no discernible features of interest.

position in our other specimens, our new schematic includes a double DMAC pattern in Lorisidae (Fig. 8B).

## Tarsiidae

In the family Tarsiidae, the lower molars are tribosphenic, with well-formed paraconid, protoconid, and metaconid cusps, and a broad lower talonid basin with entoconid and hypoconid cusps (*Schwartz, 1984*). *Swindler (2002)* describes the presence of a hypoconulid on lower third molars, but no mention of a distal cusp in M1-M2. A cristid obliqua is present, as well as a distinct buccal cingulid in all molars (*Swindler, 2002*). Thus, the current variation scheme can be summarized as a tooth crown with five primary cusps and no other cusp features (Fig. 9A). In addition to a prominent buccal cingulid that continues along the distal margin of the tooth in some of our sample, we report the presence of multiple accessory dentine horns in Tarsiidae molars. Figure 9A reveals the presence of an LMAC and BMAC in one *Tarsius spectrum* specimen. Figure 9D similarly demonstrates the presence of a BMAC in *Tarsius*, but also a BCAC close to the base of the protoconid. Figures 9E and 9F reveal patterns of multiple DMAC expression in a *Tarsius spectrum* lower first molar, and *Tarsius syrichta* lower second molar. Based on these observations, the new schematic for Tarsiidae has the addition of several accessory cusp features (Fig. 9B).

## Ceboidea

In the subfamily Callitrichinae, the lower molars have four cusps, with the mesial primary cusps connected by a crest that separates a small trigonid from a much larger talonid basin. The only notable mention of additional features in this clade is the presence of buccal cingulids on the first and second molars in most taxa except Callimico (*Kinzey, 1973*; *Swindler, 2002*). Thus, the current variation schematic can be summarized as four cusped tooth (Fig. 10A). While our observations match previous comments regarding the common presence of buccal cingulids in Callitrichinae, we extend this description by demonstrating that in some cases, these buccal features may express themselves as small dentine horns along the buccal cingulum. Where present, we identify these structures as BCACs (Fig. 10C). In addition to the buccal features presented, we also reveal the presence of multiple LMACs on the lingual marginal ridge of a *Leontopithecus rosalia* lower second molar (Fig. 10D). *Saguinas* and *Callithrix* specimens in the sample had no discernible crown features of interest (Figs. 10E and 10F). Based on these observations, the new schematic has the addition of LMACs and BCACs (Fig. 10B).

Cebinae lower molars have four cusps, with a prominent crest that separates the taller trigonid from the lower positioned talonid basin. *Swindler (2002)* states that the hypoconulid is absent in this clade, as is any form of lingual cingulid. Buccal cingulids are however reported as being variably expressed in all molars (*Kinzey, 1973*; *Orlosky, 1973*). The current variation schematic is depicted as a simple tooth crown with the four primary cusps (Fig. 11A). Contrary to Swinder's comment's however, we identify the presence of what *Swindler (2002)* would consider a hypoconulid (or more accurately, a DMAC) in a *Sapajus apella* lower first molar (Fig. 11C). Although there was no discernible lingual cingulid in our sample, the same *Sapajus apella* specimen also exhibited a small dentine

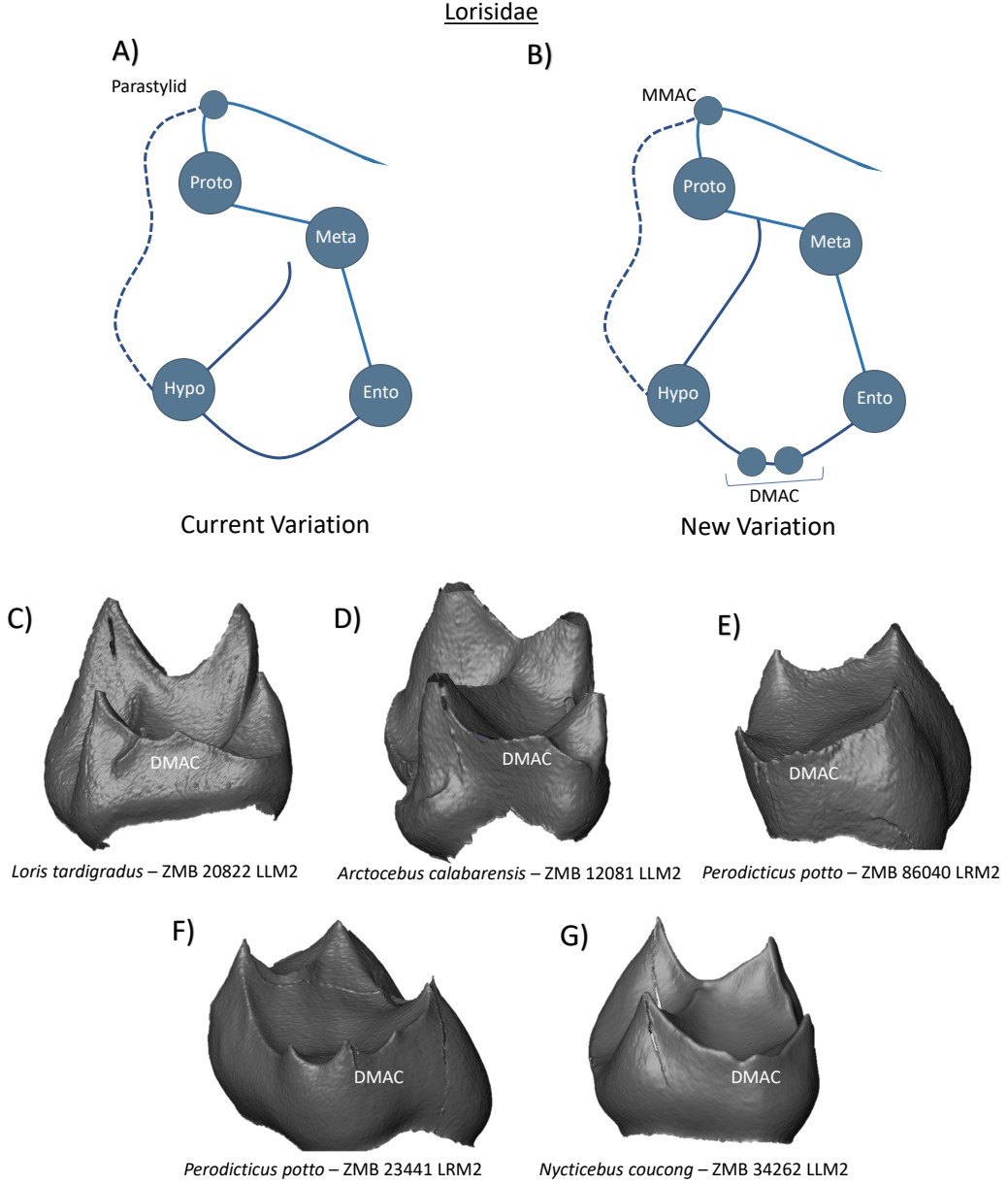

Figure 8 **Crown patterning in Lorisidae.** (A) Current variation schematic for Lorisidae, based on a review of the published literature. (B) New variation schematic for Lorisidae, based on observations at the enamel-dentine junction. (C) *Loris tardigradus* lower second molar with DMAC expression. (D) *Arctocebus calabarensis* lower second molar with DMAC expression. (E) *Perodicticus potto* lower second molar with multiple DMAC expression. (F) *Perodicticus potto* lower second molar with multiple DMAC expression. (G) *Nycticebus coucong* lower second molar with DMAC expression.

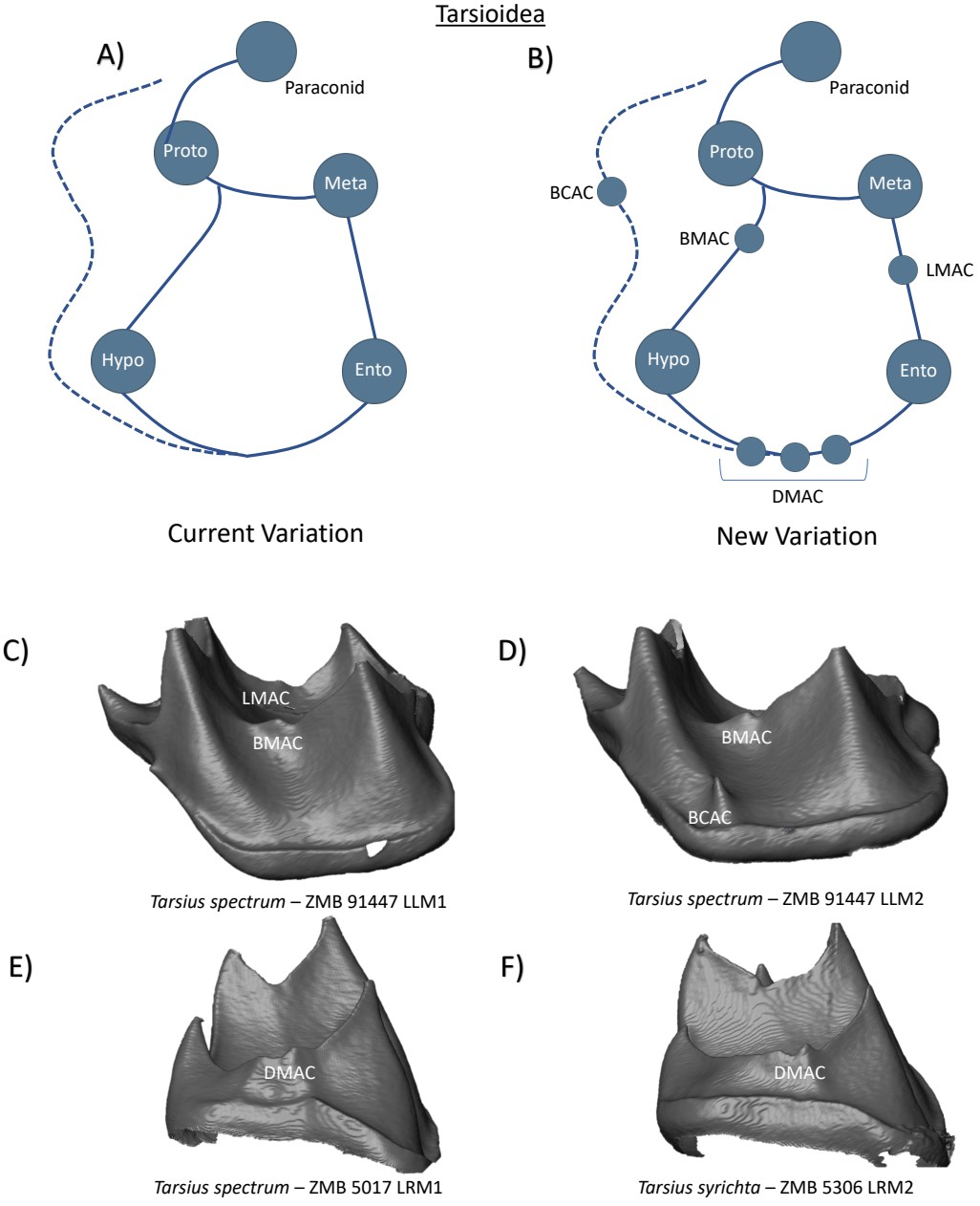

**Figure 9 Crown patterning in Tarsioidea.** (A) Current variation schematic for Tarsioidea, based on a review of the published literature. (B) New variation schematic for Tarsioidea, based on observations at the enamel-dentine junction. (C) *Tarsius spectrum* lower first molar with LMAC and BMAC expression. (D) *Tarsius spectrum* lower second molar with BMAC and BCAC expression. (E) *Tarsius spectrum* lower first molar with DMAC expression. (F) *Tarsius syrichta* lower second molar with DMAC expression.

horn on the outer slope of the metaconid. While this could be identified as a LCAC, lingual cingulid cusps were not observed in other specimens and therefore we tentatively attribute this to developmental abnormality (Fig. 11D). Regarding buccal cingulid expression, we corroborate the comments regarding buccal cingulid expression in this group and again

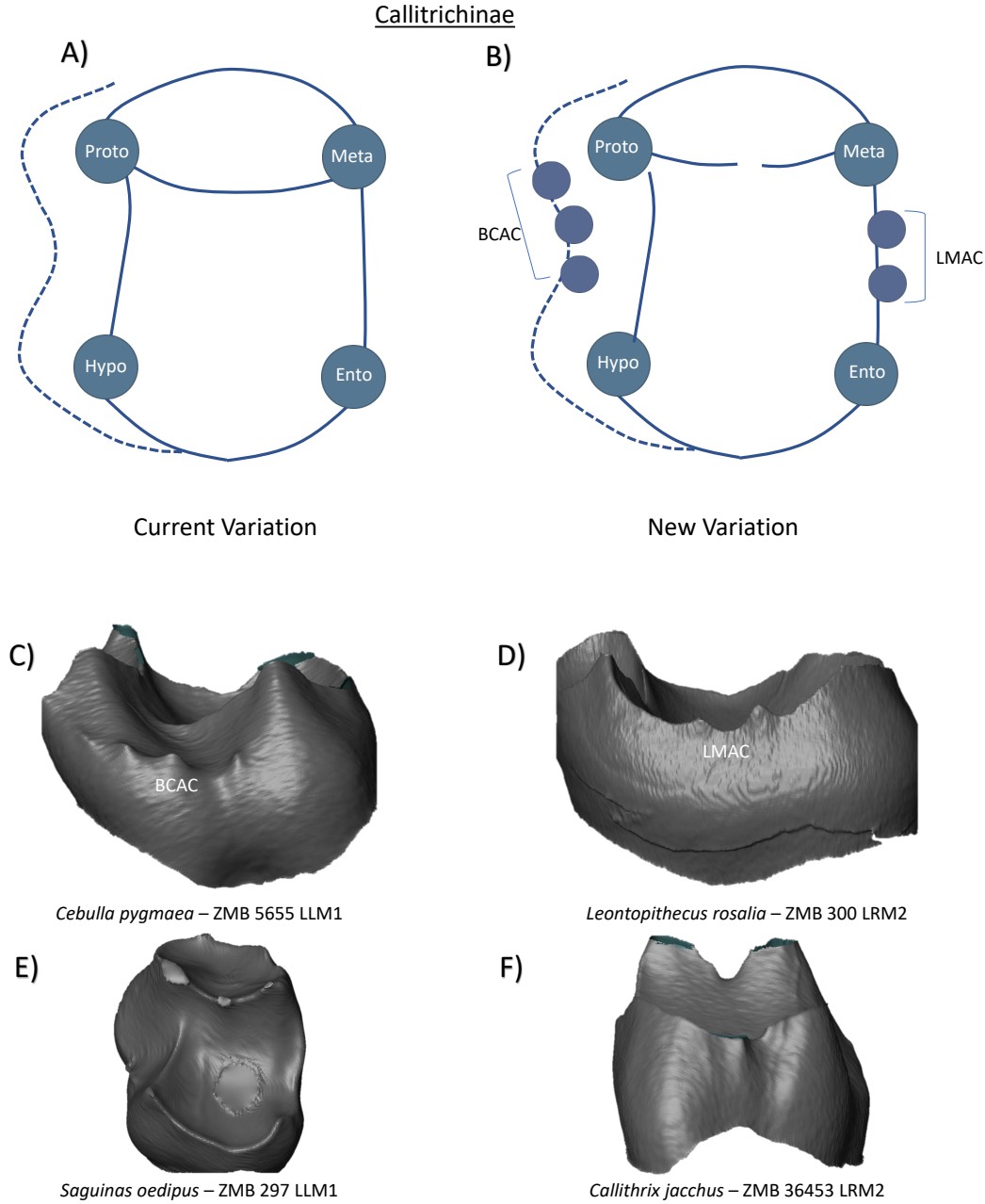

Figure 10 **Crown patterning in Callitrichinae.** (A) Current variation schematic for Callitrichinae, based on a review of the published literature. (B) New variation schematic for Callitrichinae, based on observations at the enamel-dentine junction. (C) *Cebulla pygmaea* lower first molar with BCAC expression. (D) *Leontopithecus rosalia* lower second molar with double LMAC expression. (E) *Saguinas oedipus* lower first molar with no discernible features of interest. (F) *Callithrix jacchus* lower second molar with no discernible features of interest.

identify the presence of a dentine horn (or BCAC) along the buccal cingulum in a *Saimiri sp.* specimen (Fig. 11E). Figure 11B illustrates the new schematic that we consider more appropriate and applicable to the cusp configuration of Cebinae lower molars.

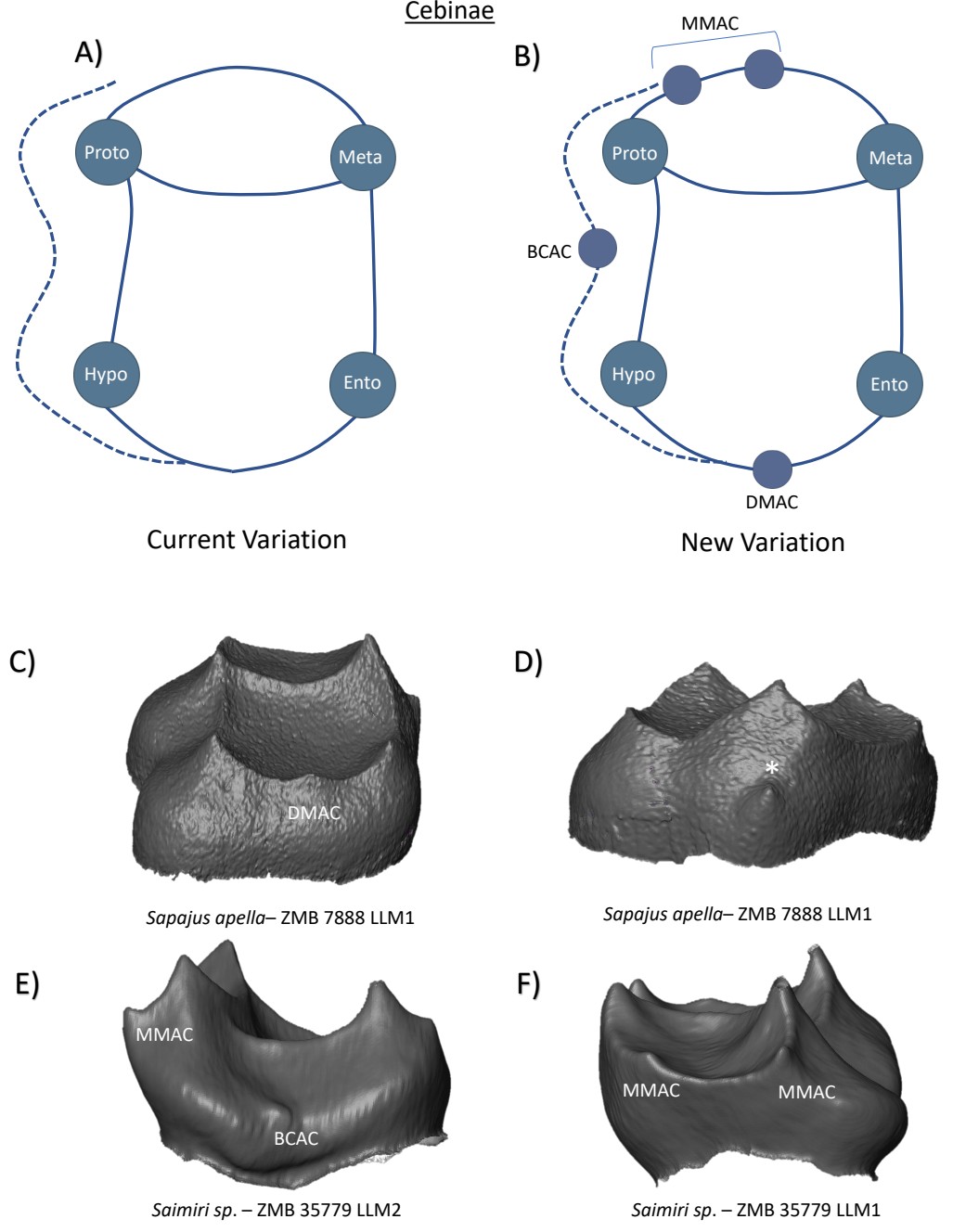

**Figure 11 Crown patterning in Cebinae.** (A) Current variation schematic for Cebinae, based on a review of the published literature. (B) New variation schematic for Cebinae, based on observations at the enamel-dentine junction. (C) *Sapajus apella* lower first molar with DMAC expression. (D) *Sapajus apella* lower first molar with developmental abnormality resembling a possible LCAC. (E) *Saimiri sp.* lower second molar with MMAC and BCAC expression (F) *Saimiri sp.* lower first molar with double MMAC expression.

In the subfamily Pitheciinae, all molars are commonly stated as possessing four cusps, with a crest connecting the protoconid and metaconid that creates a comparatively narrow trigonid and spacious taloned basin (*Swindler, 2002*). Very little is mentioned of any additional cusp-like structures in this clade, which may reflect the difficulties presented in identifying dental crown feature at the outer enamel surface in taxa with short cusps and crenulated enamel. As such, the current variation schematic is depicted as a tooth with only the four primary cusps (Fig. 12A). Our observations at the EDJ however identify the frequent presence of accessory cusps on the marginal ridges of Pitheciinae first and second molars. Figures 12C and 12D demonstrates the presence of DMAC and LMAC features in one *Cacajao calvus* specimen, while Fig. 12E provides evidence of multiple BMAC cusps on the buccal marginal ridge of a *Chiropotes satanas* lower first molar. In all Pitheciinae specimens, we identify the presence of an accessory cusp directly mesial to the protoconid (Fig. 12F). Unlike the diminutive nature of most accessory features, this cusp is often comparable in size to the neighbouring protoconid and may be mistaken for the primary cusp in some cases. While we classify this feature as an MMAC in the schematic as it is positioned on the marginal ridge between the protoconid and metaconid, further mention and consideration of this cusp will be made in the discussion. Figure 12B represents a new schematic that we consider more appropriate and applicable to the cusp configuration of Pitheciinae lower molars.

The Callicebinae subfamily in this study is represented by a small number of *Callicebus molock* specimens, but nevertheless demonstrates the presence of numerous accessory cusp features at the EDJ surface. First and second molars are reported to have four cusps, a small trigonid basin, a larger talonid basin, and a well-defined cristid obliqua (*Swindler, 2002*). In a sample of 40 *Callicebus torquatus* specimens, a 'distostylid' was identified with a frequency of 56% in lower M1 and 83% in lower M2 (*Kinzey, 1973*), while *Swindler (2002)* also remarks on the presence of this cusp in a sample of four *Callicebus moloch*. Distostylids are therefore included in the current variation schematic for Callicebinae (Fig. 13A). Our observations at the EDJ point to the presence of several DMACs on the distal marginal ridge (Figs. 13E and 13G). Additionally, we identify the presence of several MMACs on the mesial ridge (Fig. 13C), and a LMAC on the lingual ridge of a lower second molar (Fig. 13F). Finally, we demonstrate the presence of a prominent BCAC on the same second molar (Fig. 13D). Figure 13B presents the new schematic of Callicebinae lower molars that we consider to represent crown patterning in this clade.

In the subfamily Atelinae, lower first and second molars have four cusps, with a prominent crest that separates the trigonid basin from a wide talonid basin. In *Ateles*, *Orlosky (1973)* identifies the presence of hypoconulids in all lower molars. In *Alouatta* third molars, *Swindler (2002)* reports the regular appearance of hypoconulid and *tuberculum intermedium* cusps, however there is no mention of these features in first and second molars. *Clark (1971)* also described the presence of paraconid cusps on the lower molars of *Alouatta*. Incorporating these observations, the current variation scheme can be summarized as a tooth crown with four primary cusps, a hypoconulid, and a potential paraconid (Fig. 14A). Our observations at the EDJ confirm the variable presence of DMACs on the distal ridge of *Ateles* and *Alouatta* (Figs. 14C and 14F, respectively). No MMAC or LMAC were observed

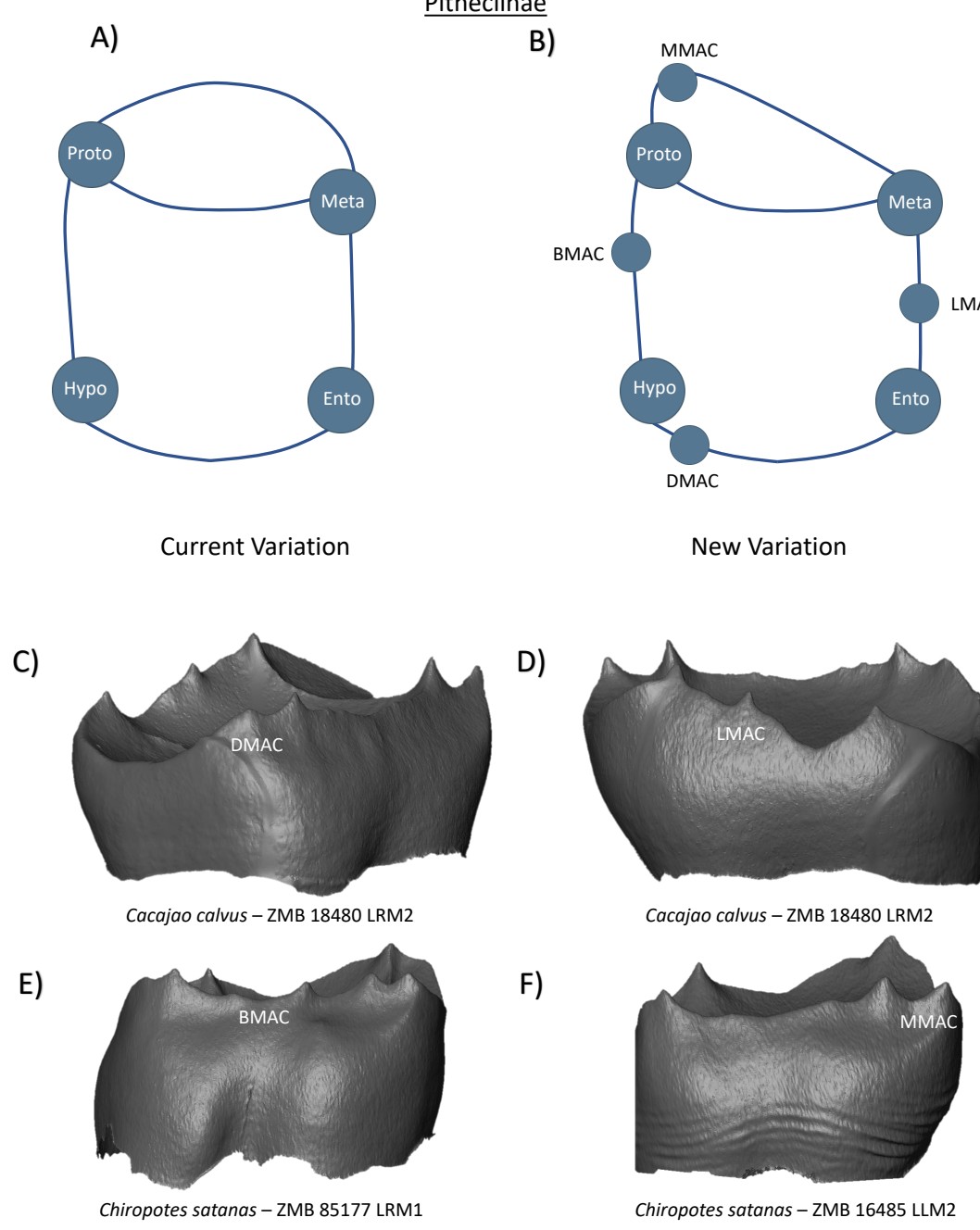

Figure 12 **Crown patterning in Pitheciinae.** (A) Current variation schematic for Pitheciinae, based on a review of the published literature. (B) New variation schematic for Pitheciinae, based on observations at the enamel-dentine junction. (C) *Cacajao calvus* lower second molar with DMAC expression. (D) *Cacajao calvus* lower second molar with LMAC expression. (E) *Chiropotes satanas* lower first molar with double BMAC expression (F) *Chiropotes satanas* lower second molar with MMAC expression.

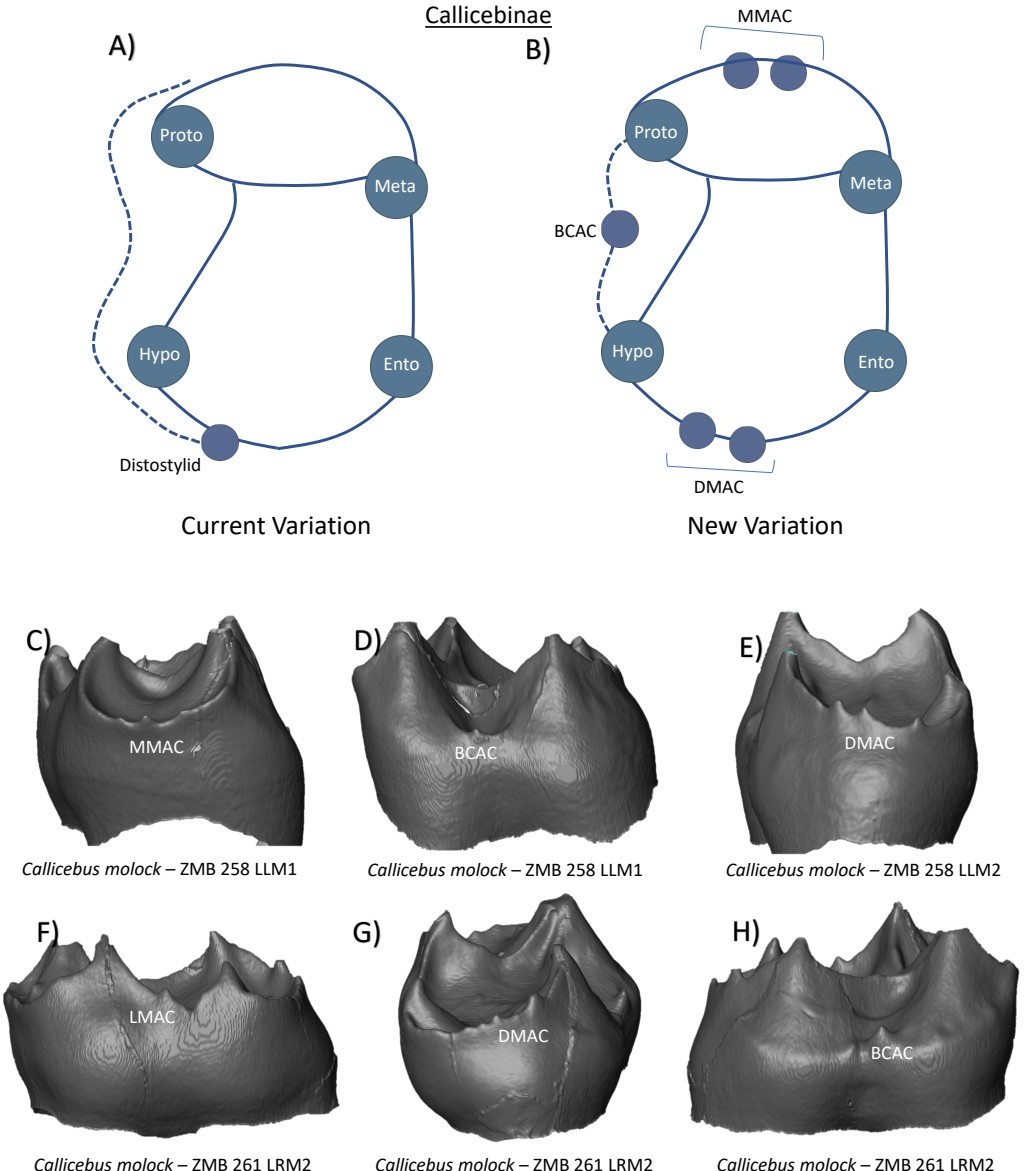

**Figure 13 Crown patterning in Callicebinae.** (A) Current variation schematic for Callicebinae, based on a review of the published literature. (B) New variation schematic for Callicebinae, based on observations at the enamel-dentine junction. (C) *Callicebus molock* lower first molar with MMAC expression. (D) *Callicebus molock* lower first molar with BCAC expression. (E) *Callicebus molock* lower second molar with multiple DMAC cusps (F) *Callicebus molock* lower second molar with LMAC expression. (G) *Callicebus molock* lower second molar with multiple DMAC cusps. (H) *Callicebus molock* lower second molar with BCAC expression.

in Atelinae first or second molars. Figure 14D demonstrates the lack of discernible features on the lingual ridge of an *Alouatta* specimen. In relation to *Clark*'s (*1971*) description of a paraconid in *Alouatta*, we find no examples of paraconid expression in our limited sample. Figure 14E does however demonstrate notable lipping and elevation of the mesial

marginal ridge in an *Alouatta* individual. As this type of ridge morphology may resemble cusp-like structures at the outer enamel surface in some specimens, we tentatively attribute the descriptions of potential paraconid expression to this phenomenon and exclude them from the new schematic for Atelinae until they have been confidently detected (Fig. 14B).

## Cercopithecidae

The two subfamilies of Cercopithecidae are presented separately, and the Cercopithecinae subfamily is further separated into their Cercopithecini and Papionini tribes, due to notable differences in molar morphology and accessory cusp expression between the groups.

In the Tribe Cercopithecini, the molars are bilophodont and have high, well-defined cusps. There are a limited number of studies on variation in crown morphology from this group but from a large sample of guenons, *Swindler (2002)* reported the lack of hypoconulid presence on all lower molars, but the common expression of a protostylid on the lower molars of *Chlorocebus aethiops* (85% of specimens) and *Miopithecus talapoin* (100% of specimens). Thus, the current variation scheme can be summarized as a relatively simple tooth crown with the four primary cusps and a protostylid located on the mesiobuccal corner of the crown (Fig. 15A). Our observations at the EDJ of guenon first and second molars correspond with *Swindler*'s (*2002*) observations regarding the lack of hypoconulid (or DMAC) presence (Figs. 15C, 15D, 15E and 15F). Whether this relates to developmental constraints associated with a notably small distal fovea in this clade, remains to be determined. Unlike *Swindler*'s (*2002*) observations in relation to protostylid expression, however, we find no evidence of a protostylid (or BCAC) in any specimen. This is consistently true, even when alternative definitions of protostylids are adopted (see *Hlusko, 2004*; *Skinner et al., 2008* for discussion of protostylid definitions). Due to the limited sample size available for this study, we cannot rule out the presence of a BCAC in some individuals and thus it is included in our new Cercopithecini schematic (Fig. 15B). Unlike the other Cercopithecidae groups, no other cusp features beyond the four main cusps were observed.

In the Tribe Papionini, molars are bilophodont and display a pronounced buccal flare. *Swindler (2002)* reports that accessory cusps are variably expressed in several members of this tribe, although are more commonly observed in *Macaca* and *Papio* molars. In *Macaca fuscata* lower second molars, *tuberculum intermedium* presence on the lingual aspect of the crown was reported in 38% of specimens, and in 56.8% of *Papio* first lower molars (*Swindler, 2002*). These features are therefore incorporated into the current variation scheme for Papionini (Fig. 16A). While our observations match Swindler's comments regarding the common presence of a *tuberculum intermedium* cusp (or LMAC) in Papionini molars, we extend this description by demonstrating the presence of multiple lingual accessory cusps on the marginal ridge between the metaconid and entoconid in some taxa (Fig. 16C). In these specimens, LMACs are often positioned either deep within the lingual fovea, or on the distal shoulder of the metaconid. Currently, no more than two LMACs have been observed on any Papionini lower M1 or M2, however the new nomenclature allows for the addition of extra cusps if observed. Regarding distal accessory cusp expression, *Swindler (2002)* comments that it is well known that a shelf or cusp extends from the distal

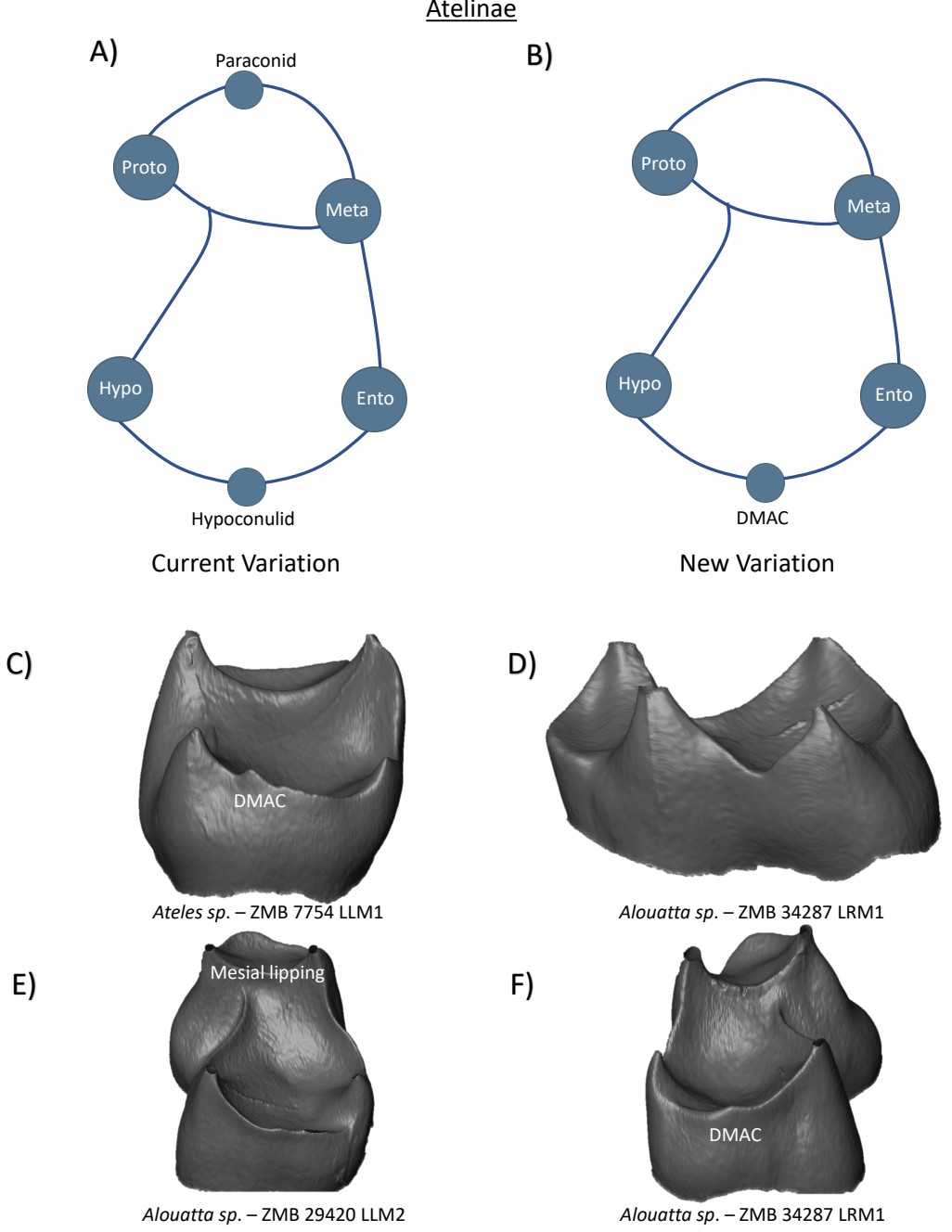

**Figure 14 Crown patterning in Atelinae.** (A) Current variation schematic for Atelinae, based on a review of the published literature. (B) New variation schematic for Atelinae, based on observations at the enamel-dentine junction. (C) *Ateles sp.* lower first molar with DMAC expression. (D) *Alouatta sp.* lower first molar with no discernible features of interest. (E) *Alouatta sp.* lower second molar with mesial marginal lipping, but no discernible accessory cusp (F) *Alouatta sp.* lower first molar with DMAC expression.

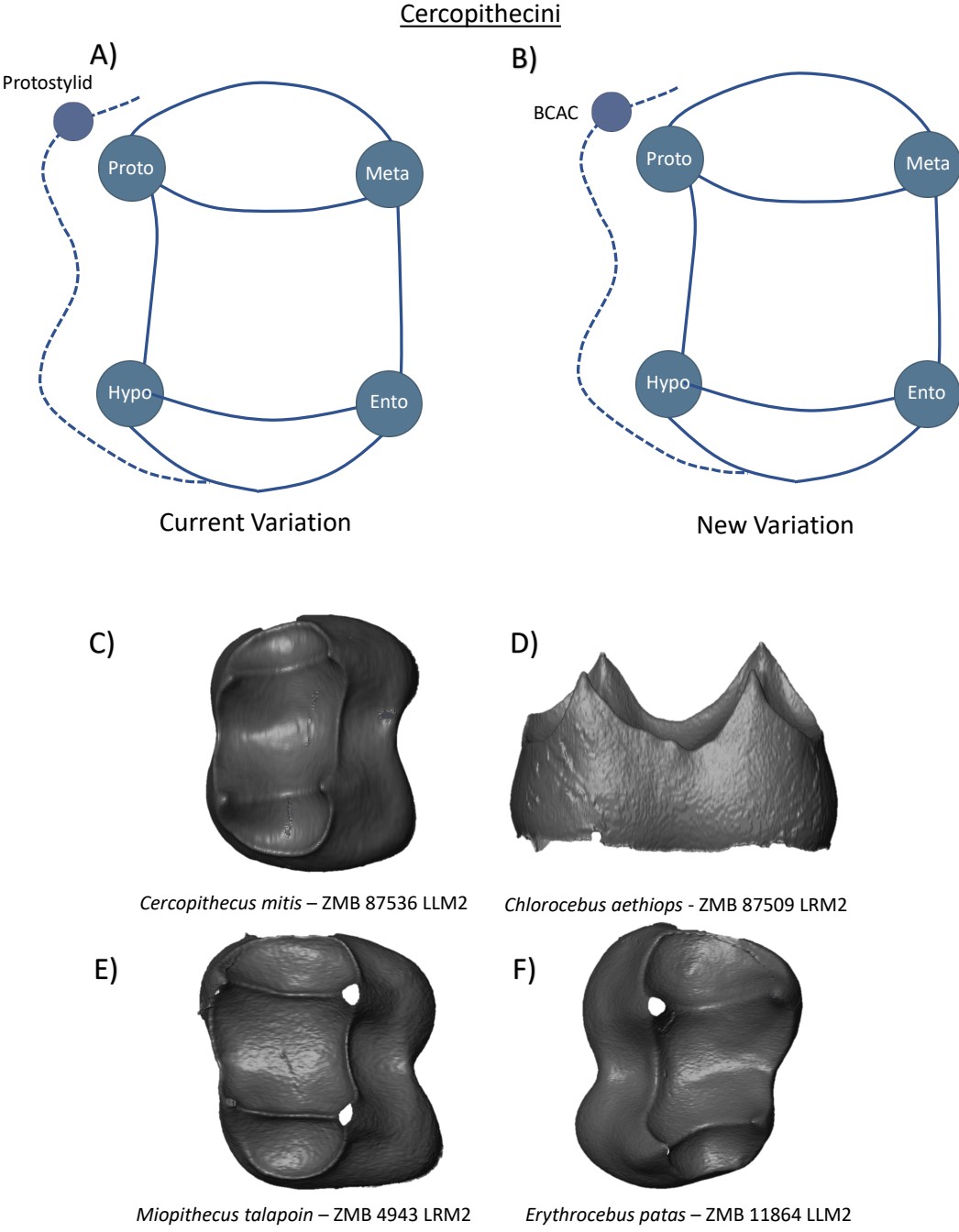

**Figure 15  Crown patterning in Cercopithecini.** (A) Current variation schematic for Cercopithecini, based on a review of the published literature. (B) New variation schematic for Cercopithecini, based on observations at the enamel-dentine junction. (C) *Cercopithecus mitis* lower second molar with no discernible features of interest. (D) *Chlorocebus aethiops* lower second molar with no discernible features of interest. (E) *Miopithecus talapoin* lower second molar with no discernible features of interest. (F) *Erythrocebus patas* lower second molar with no discernible features of interest.

surface of Papionini molars, and that in the lower M3, it is considered the hypoconulid. *Szalay & Delson (2013)* describe in *Theropithecus* molars the presence of a "large distal accessory cuspule, which projects backward towards the succeeding tooth" (p. 375). There is no indication of a name from *Szalay & Delson (2013)* regarding this structure and how they would define it; however, *Swindler (1983)* has suggested that the distal cusps on the lower M1-2 are serially homologous with the hypoconulid of the M3. It should be noted however, that this is based on topographical and functional associations, not phylogenetic. Our studies at the EDJ support the comments regarding the common observation of distal accessory cusps in Papionini molars. Developing on these descriptions, we report a variable and complex pattern of cusp expression in this clade, including the large single cuspules described by *Szalay & Delson (2013)*, as well as the common occurrence of multiple dentine horn presence along the distal ridge (Figs. 16D, 16E, and 16F). Importantly, from the images of multiple distal accessory cusp expression in this clade, we would argue that even if one wishes to label these features within the current system of nomenclature, the identification and differentiation of a 'hypoconulid' from the other cusps present could not be made with confidence. As such, we consider this to support the adoption of the term 'DMAC' for all distal accessory cusps in this clade.

In relation to cusp features beyond the cusps commonly situated on the buccal and distal aspects of the crown, *Hlusko (2002)* studied variation in 'interconulid' expression among a large sample of *Papio hamadryas,* and identified some form of expression in almost the entire collection (95%). In this case, *Hlusko (2002)* used *Swindler*'s (*1976*) definition of an 'interconulid' as a stylid between the protoconid and hypoconid of mandibular molars. As previously mentioned however, photographs provided of interconulid expression types in this study appear to show features on the buccal cingulum and perhaps better reflect what has previously been termed an 'ectostylid' (*Kinzey, 1973*). Nevertheless, similar buccal features have also been observed at the EDJ in our sample. These include cusp-like structures on the buccal marginal ridge, and on the buccal cingulum (Fig. 16H). Based on the potential confusion regarding the term interconulid, and the acknowledgement of cusp-like features on the buccal marginal ridge (which do not appear to have a suitable or commonly referenced name), we incorporate BMAC (buccal marginal accessory cusp) and BCAC (buccal cingulum accessory cusp) terms into the new nomenclature to facilitate the identification and differentiation of these features (Fig. 16B). Additionally, we also demonstrate the presence of MMACs (mesial marginal accessory cusps) on the mesial marginal ridge of Papionini molars. Similar to DMAC presence in Papionini M1-M2, patterns of MMAC expression vary from absent to multiple dentine horn expression along the marginal ridge (Fig. 16G).

In the other subfamily of Cercopithecidae, the Colobinae, there is limited discussion in the literature of any particular morphological feature on the molar crowns that may be of interest to this study. The Colobinae are described as having four cusps on the first and second molars, with a variably expressed hypoconulid on the M3. *Swindler (2002)* notes the variable presence of a *tuberculum intermedium* in *Rhinopithecus* on the lower M1 (7%) and M2 (62%), and on the M1 (9%) of *Pygathrix*. The current nomenclature for this clade can therefore be summarized as a simple tooth crown with four primary cusps, and a

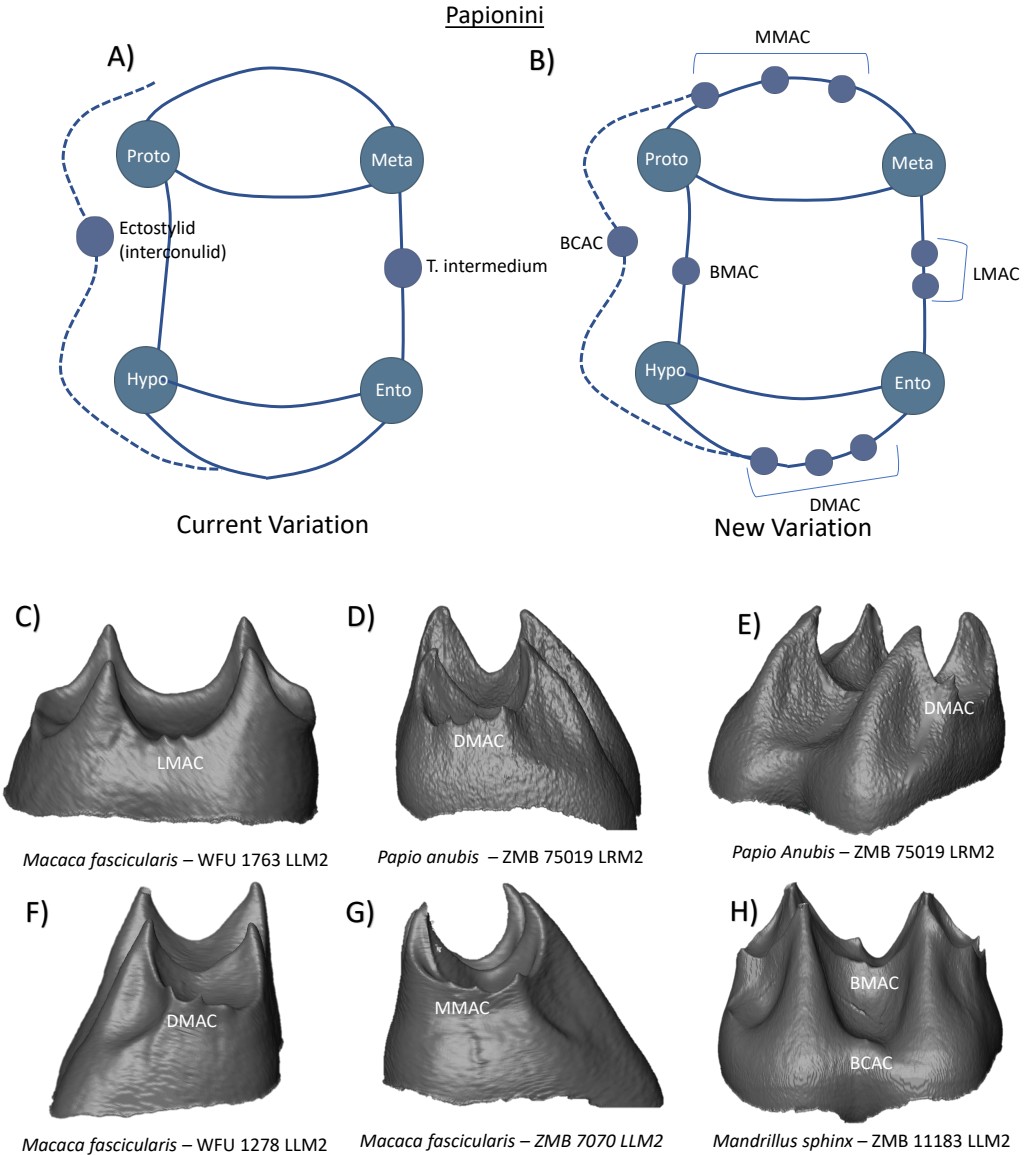

**Figure 16 Crown patterning in Papionini.** (A) Current variation schematic for Papionini, based on a review of the published literature. (B) New variation schematic for Papionini, based on observations at the enamel-dentine junction. (C) *Macaca fascicularis* lower second molar with double LMAC expression. (D) *Papio anubis* lower second molar with double DMAC expression. (E) *Papio anubis* lower second molar with DMAC expression. (F) *Macaca fascicularis* lower second molar with DMAC expression. (G) *Macaca fascicularis* lower first molar with double MMAC expression. (H) *Mandrillus sphinx* lower second molar with BMAC and BCAC expression.

potential C7 (aka LMAC) on the lingual marginal ridge (Fig. 17A). While *Rhinopithecus* is not included in our sample, we see no examples of what we would consider a LMAC in any of our Colobinae specimens. Nevertheless, due to low sample sizes among some groups, and incomplete representation of the whole subfamily of Colobinae, we cannot rule out the presence of lingual accessory cusps in some individuals, and would recommend the

LMAC designation for these features in future studies. Figure 17C shows the presence of a DMAC in a *Colobus guereza* lower right second molar, and a DMAC in two *Presbytis melaophos* molars (Figs. 17D and 17E). Specimens F–H in Fig. 17 demonstrate the simple M1/M2 molar morphology in this clade, and lack of discernible accessory features within these specimens. Based on these findings, the new schematic has the inclusion of both LMAC and DMAC (Fig. 17B).

## Hominoidea

In the family Hominidae, molars generally have five main cusps, arranged in a Y-5 pattern. Importantly, because the fifth cusp (or hypoconulid) is consistently expressed in M1-M2 in this clade, it is not considered to be an accessory feature and is included as a 'hypoconulid' in the following schematics. Further dialogue regarding the inclusion of the term hypoconulid in this clade follows in the discussion. In addition to these five cusps, the variable presence of a *tuberculum sextum* and *tuberculum intermedium* are commonly reported in certain members of this clade. *Swindler (2002)* reports that *Gorilla* have the highest frequency of C6, while *Pongo* have the least. Previous studies of dental trait expression at the EDJ have provided a more detailed analysis of C6 and C7 expression in hominoids, noting variation in the placement of the accessory dentine horns on their respective marginal ridges, as well as observations of double C6 in some Homininae specimens (*Skinner et al., 2008*). The current variation schematic can therefore be summarized as a tooth crown with four primary cusps, a variable C7 cusp, and multiple potential C6 cusps (Fig. 18A). Our observations corroborate the presence of multiple DMACs on the distal ridge between the hypoconulid and entoconid of some Homininae molars (Figs. 18C and 18D). Assuming that the larger of the two distal cusps is the hypoconulid, Fig. 18D demonstrates the rare presence of a DMAC between the hypoconulid and hypoconid. While variation in the patterning and placement of these cusps may reflect developmental differences in their formation and expression, we still consider the use of the term DMAC for all distal accessory cusps appropriate as it does not intend to imply homology. In addition to DMAC expression, we also note the presence of LMACs in several of our Hominidae sample. Figs. 18E and 18F demonstrate the presence of single LMACs in a *Homo sapiens* and *Gorilla gorilla* specimen respectively. No other accessory features were observed. As such, the new schematic of Hominidae lower molars does not include any new features but does replace C6 and C7 terms with the more appropriate DMAC and LMAC designations (Fig. 18B).

In the family Hylobatidae, the lower molars possess five cusps, a narrow trigonid, and a more spacious talonid basin. Reports of a Y-5 pattern follow frequencies of roughly 100% in LM1, and 97% in LM2 (*Frisch, 1965*; *Swindler, 2002*). *Swindler (2002)* reports *tuberculum intermedium* expression of 0.07% in LM1 and 18% in LM2 in a sample of *Hylobates* molars. Tuberculum sextum presence, or any other form of accessory trait expression, was not discussed. The current variation schematic can be summarized as a crown with four primary cusps, a prominent hypoconulid, and a potential *tuberculum intermedium* (Fig. 19A). From our observations, we identify the presence of a small DMAC between the hypoconulid and entoconid in a *Hylobates muelleri* M2 (Fig. 19C), and a single example of a very mesially-positioned LMAC in *Hylobates muelleri* M1 (Fig. 19D). No

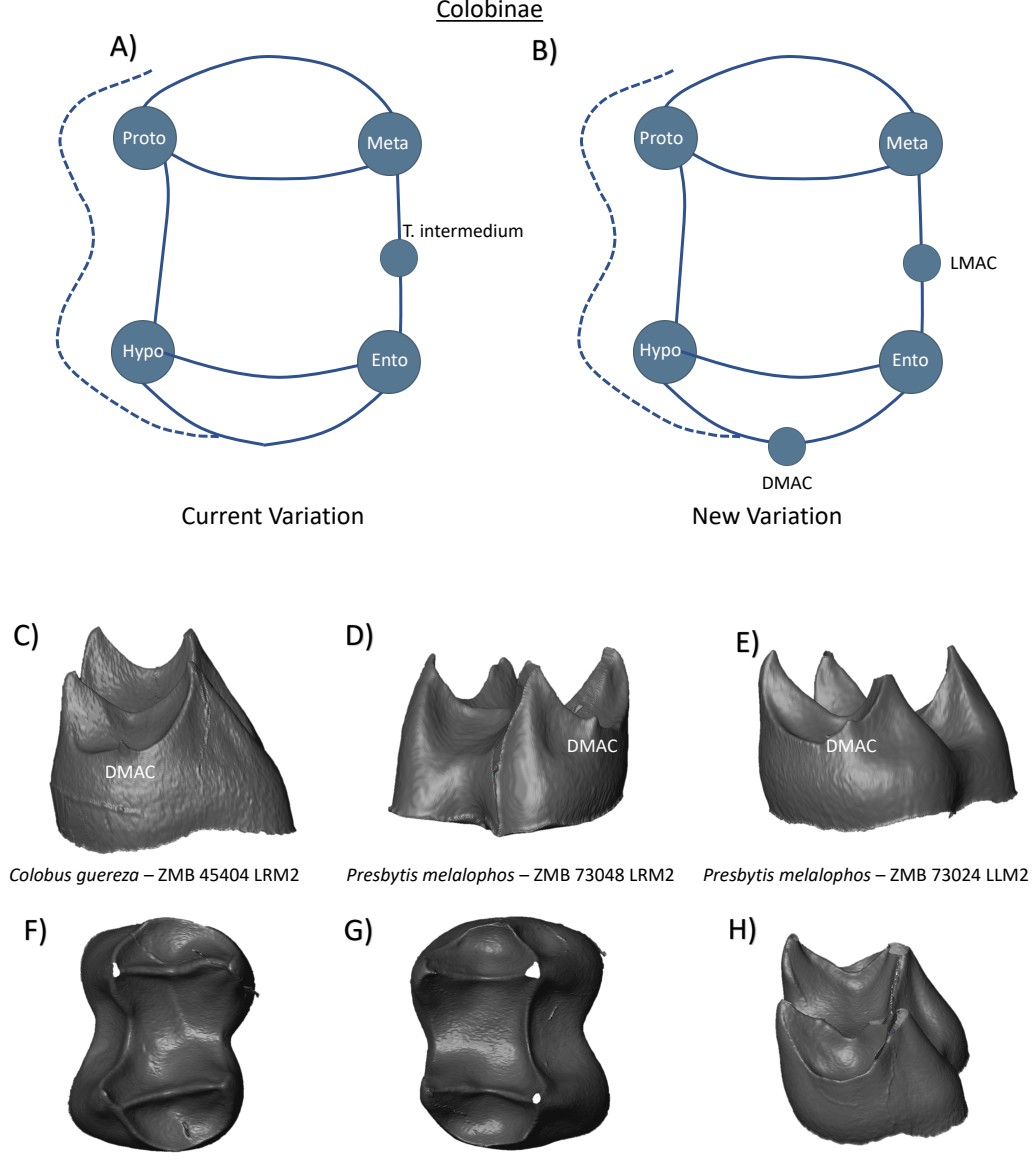

**Figure 17 Crown patterning in Colobinae.** (A) Current variation schematic for Colobinae, based on a review of the published literature. (B) New variation schematic for Colobinae, based on observations at the enamel-dentine junction. (C) *Colobus guereza* lower second molar with DMAC expression. (D) *Presbytis melalophos* lower second molar with DMAC expression. (E) *Presbytis melalophos* lower second molar with DMAC expression (F) *Nasalis larvatus* lower second molar with no discernible features of interest. (G) *Trachypithecus cristatus* second first molar with no discernible features of interest. (H) *Trachypithecus cristatus* lower second molar with no discernible features of interest.

dentine horns beyond the five primary cusps were observed in our *Symphalangus* sample (Figs. 19E and 19F). Figure 19B represents the new schematic of Hylobatidae lower molars that we consider to represent a better reflection of crown patterning in this clade.

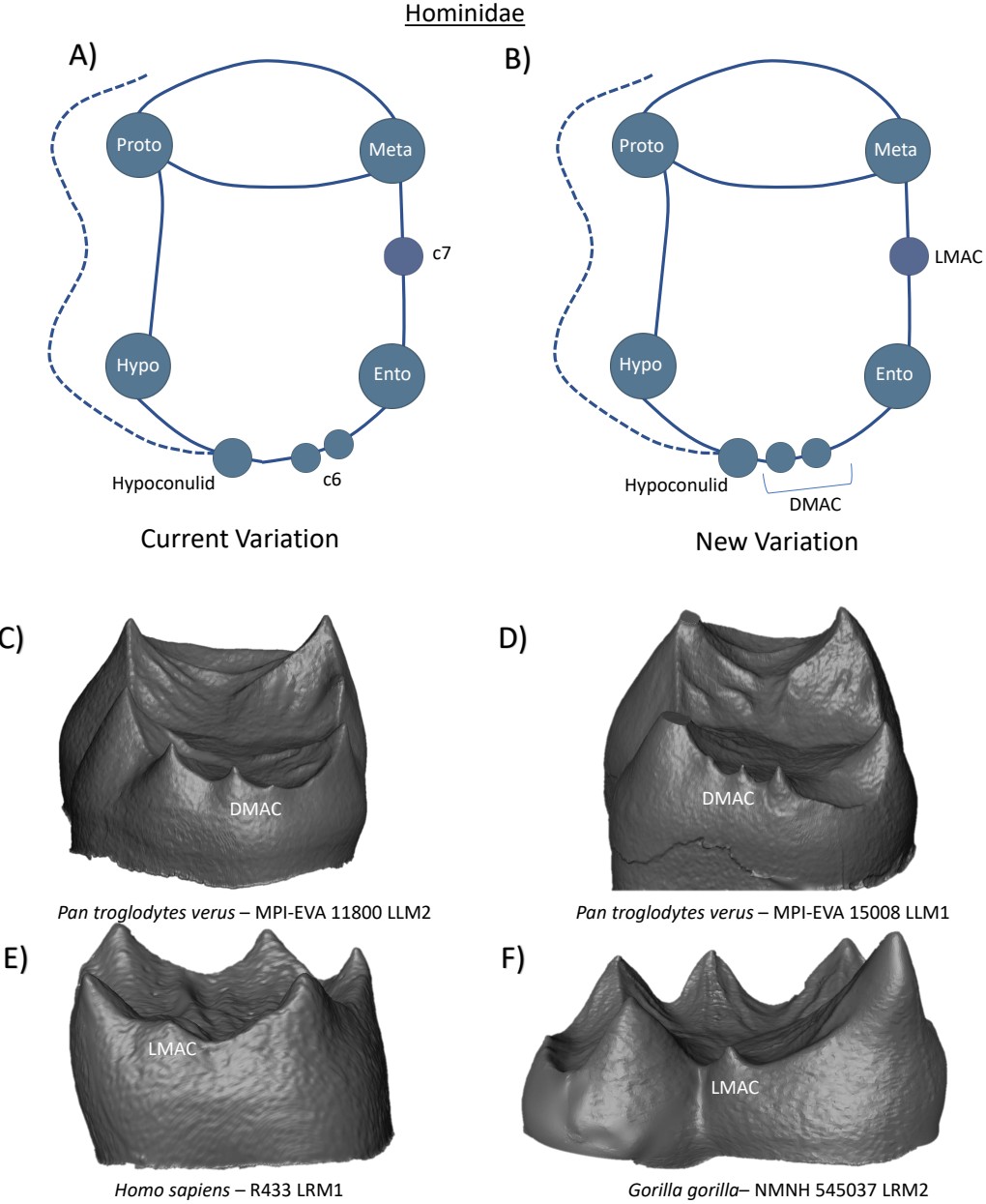

**Figure 18 Crown patterning in Hominidae.** (A) Current variation schematic for Hominidae, based on a review of the published literature. (B) New variation schematic for Hominidae, based on observations at the enamel-dentine junction. (C) *Pan troglodytes verus* lower second molar with double DMAC expression. (D) *Pan troglodytes verus* lower first molar with double DMAC expression. (E) *Homo sapiens* lower first molar with LMAC expression. (F) *Gorilla gorilla* lower second molar with LMAC expression.

## DISCUSSION

Our broad study of primate EDJ morphology demonstrates the presence of numerous dental crown features that either have not been previously observed, have not been previously identified in their respective taxa or clade, or display a greater level of variation and

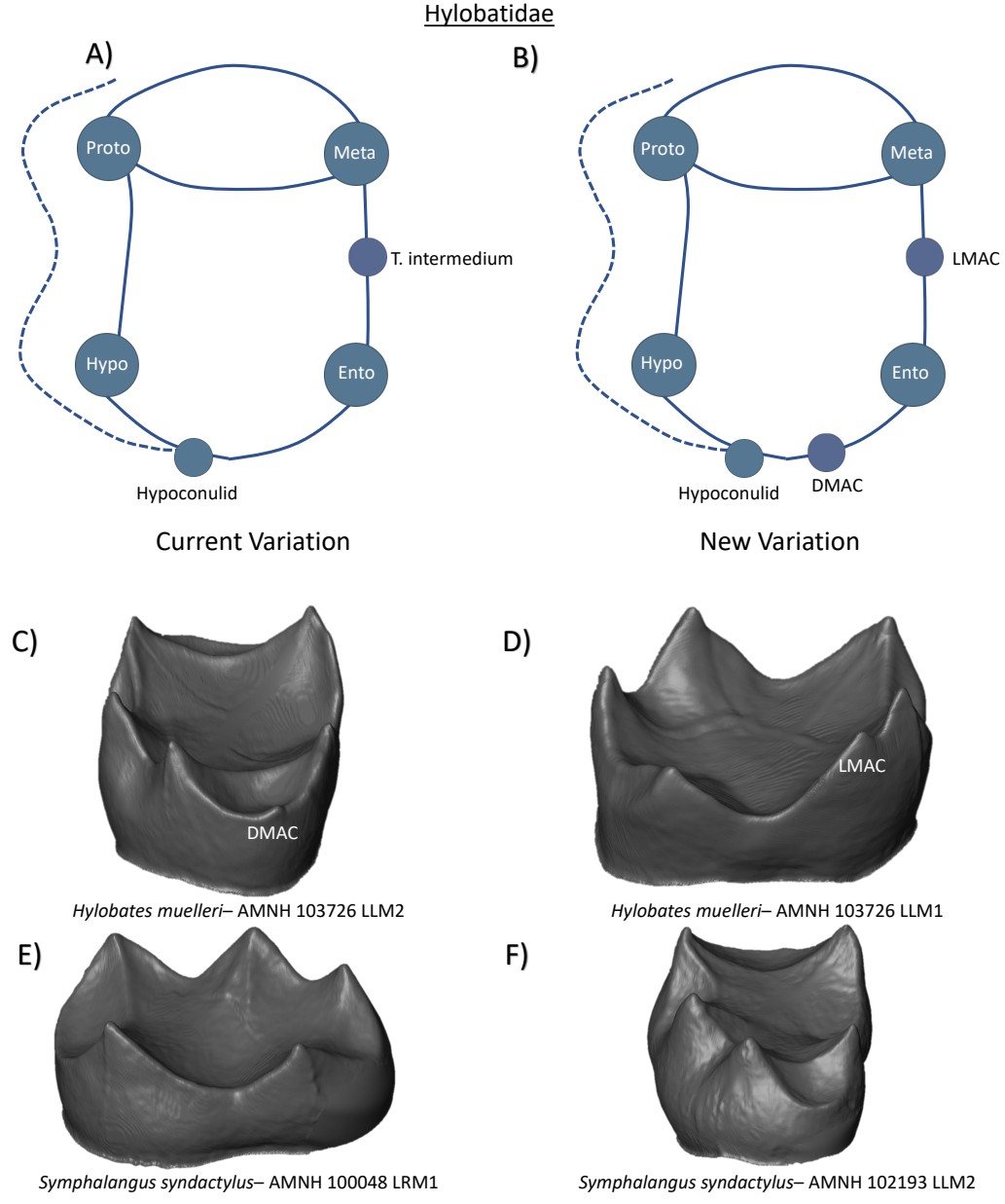

**Figure 19 Crown patterning in Hylobatidae.** (A) Current variation schematic for Hylobatidae, based on a review of the published literature. (B) New variation schematic for Hylobatidae, based on observations at the enamel-dentine junction. (C) *Hylobates muelleri* lower second molar with DMAC expression. (D) *Hylobates muelleri* lower first molar with LMAC expression. (E) *Symphalangus syndactylus* lower first molar with no discernible features of interest. (F) *Symphalangus syndactylus* lower second molar with no discernible features of interest.

complexity than has previously been assumed. As many of these features present further complications and uncertainties to an already challenging and sometimes misleading system of nomenclature, we introduce and discuss each accessory cusp within a proposed system that focuses on a simple location-based categorisation (*e.g.,* BCAC, LMAC, *etc.*).

Until there is a better understanding of the developmental origin, evolutionary history, and forms of variation and expression of these features in each clade, we consider this system to be the most practical way of discussing these structures. Previously, dental morphologists have conceded to naming accessory features within one of the current entrenched systems of nomenclature, despite an awareness of its potential unsuitability. While intentionally void of homological, evolutionary, and developmental information, the system proposed here allows for the clear identification and positioning of crown features, without using a term that is historically or developmentally loaded. As micro-CT scanning and observations of EDJ morphology in primates continue, it is hoped that a better understanding of the various forms of trait expression in each clade can be understood.

In addition to the acknowledgement of novel cusp patterning in numerous primate groups, this study also emphasises the importance of why a single nomenclature or diagram for all primate molars is impractical. While there is an obvious appeal to the establishment and utilisation of a single nomenclature, the extent of the variation in cusp presence and expression shown in this study demonstrates how this is not possible or justifiable. Based on our limited understanding of the development and phylogenetic history of many of these features at this time, the creation of a single diagram or schematic for all primate molars requires either (1) the inclusion and separation of all crown features seen across all primate groups, creating a densely inhabited collection of individual features that would commonly repeat equivalent structures, (2) or attempt to reduce all observable variation down to a small number of crown features that are topographically similar, which would grossly overlook the subtle but unique crown patterning seen in certain clades. As such, we consider clade-specific diagrams to be the most logical solution. Clade-specific diagrams provided in this study group taxa of similar cusp patterning by the highest taxonomic rank possible, including clades at the family, subfamily, and tribe rank. As molars from additional members of these groups are observed and a deeper understanding of trait expression in each clade is gained, these diagrams may need to adjust their taxonomic rank to accurately reflect and distinguish between new patterns of expression among closely-related taxa.

While the proposed terms and clade-specific diagrams provided here suggest a complete departure from all previous systems of nomenclature for accessory cusps, and thereby a departure from any system that attempts to infer homology or phylogenetic relation, this is not the recommendation for all future work. Currently, based on the variation demonstrated here, we advise the use of the location-based categorisation introduced here as an alternative to the variously flawed current terms. However, when a better understanding of the development and history of a particular crown feature is known, we encourage the reintroduction or establishment of new, more appropriate terms. We advise doing so only if the individual cusp can be consistently and clearly identified to the exclusion of other nearby cusps, is consistently expressed in most/all members of a specific group, and can be identified and tracked through early representative of the clade. Importantly, these new terms should be exclusive to their respective group, and not used to describe similar features in phylogenetically distant taxa. From observations of a large sample of Hominoidea lower molars, we consider the large cusp of the distal marginal ridge to match these criteria and have thus attributed it to the term 'hypoconulid'. Similar

(re)introductions may also be appropriate for the accessory cusp directly mesial to the protoconid in Pitheciinae molars, however small sample sizes restrict us from making more definitive assertions.

Based on a need for simplicity and an insufficient understanding of the precise developmental processes responsible for accessory cusp formation, it is sensible at this time to attribute all cusps to the simple location-based categorisations we have provided. However, the expression type and positioning of many cusp features hint at potential variations in the developmental processes or contributory factors responsible for certain cusp patterns that may warrant the introduction of individual and more suitable terms as this variation is understood. For example, while all cusps on the lingual marginal ridge are labelled as LMACs and are often positioned deep within the fovea between the metaconid and entoconid, they also commonly occur on the distal slope of the metaconid (see Hylobates muelleri specimen AMNH 103726 in Fig. 19D). *Skinner et al. (2008)* and *Davies et al. (2021)* have previously identified different forms of C6 and C7 expression and positioning along the lingual and distal margins of hominoid molars, and differentiated them accordingly in their schematics and discussion. Potential complications regarding cusp expression types and positioning are further exaggerated with the acknowledgement and inclusion of cusp shouldering features. The current schematics recognise only clearly identifiable cusps, with an elevated dentine horn relative to all corresponding sides of the horn tip. However, there are several examples of cusp shouldering in primate molars, in which the shoulder is present only as a convexity of the marginal ridge close to a larger cusp (see Galago senegalensis ZMB 64278 in Fig. 15C).

Currently, it is unclear whether cusp shouldering is developmentally homologous to a dentine horn and should be considered equivalent to the minor expression of an accessory cusp. Similar issues regarding the potential distinction between cusps and crest-like features are also present at the cingulum. For example, while some recognise a protostylid as a cusp found on the buccal surface of the protoconid (*Turner, Nichol & Scott, 1991*; *Hlusko, 2004*), others have suggested that an elevation or ridge on the anterior part of the buccal surface may be the product of the same developmental process and thus should also be considered within protostylid expression (*Skinner et al., 2008*; *Skinner, Wood & Hublin, 2009*). As there is limited understanding of the developmental processes responsible for crest patterning, and the focus of this study was regarding cusp patterns specifically, non-cusp related cingulid features were not included in the schematics. While the observations in this study recognise several different forms of cusp expression in primates that may relate to important differences in the development and growth of these features, attempts to distinguish between these cusp patterns in the schematics was avoided. While it may be useful to separate these expression types in some cases, the overwhelming degree of variation that exists in individual cusp positioning in primates would introduce numerous inconsistencies regarding the confident categorisation of expression types. Furthermore, the introduction of separate terms that in any way imply that these features are developmentally distinct would not be appropriate at this time.

While this study focuses on the crown morphology of primate mandibular first and second molars, novel patterns of cusp expression are also present in other tooth positions.

Due to the significant cost and effort associated with micro-CT scanning and image processing, this study focused only on mandibular first and second molars, preferring a broad sample of taxa than of tooth position. It is highly likely however, that similar patterns of cusp expression will be present on upper molars. Additionally, the intentional exclusion of third molars from this study partially reflects the high degree of variability and expression at the crown surface, that is often notably exaggerated or restrained relative to the other molars. For many taxa, third molars will require their own schematic diagrams, and involve difficult decisions regarding the confident identification of features. For example, certain members of Papionini and Colobinae clades exhibit multiple large cusps along the distal marginal ridge of lower third molars that would currently be extremely difficult to accurately identify and differentiate. While it will be important to decide whether morphologically similar anatomical structures on third molars are homologous with those on first and second molars, serial homology and the application of equivalent terms will also be relevant to studies of premolar morphology. Finally, we expect that similar challenges presented here also exist for the analysis of upper dentitions, and will likely require the introduction of similar schematics and categorisations.

## CONCLUSIONS

In this study, we reveal new patterns of lower molar accessory cusp expression in primates. In particular, we highlight the numerous discrepancies between the expected patterns of variation inferred from the current academic literature, and the new patterns of expected variation seen in this study. This new variation includes the presence of dental crown features that have not been previously observed or reported in any primate taxa. In other cases, we extend previous observations of crown structures in certain primate groups to new primate clades or taxa. Importantly in the majority of cases, we do not consider these latter observations to be homologous with their original reporting. As such, rather than attempting to label these features within one of the previously used and established systems of nomenclature, we introduce each feature within a conservative, non-homologous scheme that focuses on simple location-based categorisations. Until there is a better insight into the developmental processes and phylogenetic history associated with each individual feature in each clade, these categorisations are the most practical way of addressing these structures. As an understanding of the development and evolutionary history of crown features improves, we encourage the establishment of more appropriate, informative, clade-specific terms.

## ACKNOWLEDGEMENTS

Access to specimens was kindly provided by the following institutions: Museum für Naturkunde, Leibniz-Institut für Evolutions- und Biodiversitätsforschung an der Humboldt-Universität zu Berlin, Wake Forest University Primate Centre, Kyoto University Primate Research Institute, American Museum of Natural History in New York, Musee Royal de l'Afrique Centrale in Tervuren, Max Planck Institute for Evolutionary Anthropology, Institutul de Antropologie ''Francisc J. Rainer'', Universität Leipzig, Institut

 

für Anatomie, National Museum of Natural History in Washington, Institut für Anatomie und Zellbiologie in Greifswald, and Forschungsinstitut Senckenberg in Frankfurt. For access to specimens we thank Emmanuel Gilissen and Wim Wendelen, Ottmar Kullmer and Friedemann Schrenk, Frieder Mayer, Christophe Boesch, the Ministry of Environment and Eaux et Forêts, the Ministry of Scientific Research, the Direction of the Taï National Park as well as the Swiss Centre of Scientific Research. For scanning assistance we thank Heiko Temming, Andreas Winzer, Patrick Schönfeld, Anthony Olejniczak, Robin Feeney, and Tanya Smith. Macaque dentitions were kindly loaned by Ellen Miller and the Wake Forest University Primate Centre. We thank Akiko Kato, and Nancy Tang for scanning assistance at the Harvard University Centre for Nanoscale Systems (CNS).

### Funding

This project has received funding from the European Research Council (ERC) under the European Union's Horizon 2020 research and innovation programme (grant agreement No. 819960). The funders had no role in study design, data collection and analysis, decision to publish, or preparation of the manuscript.

### Grant Disclosures

The following grant information was disclosed by the authors:
European Research Council (ERC) under the European Union's Horizon 2020 research and innovation programm: 819960.

### Competing Interests

The authors declare there are no competing interests.

### Author Contributions

- Simon A. Chapple conceived and designed the experiments, performed the experiments, analyzed the data, prepared figures and/or tables, authored or reviewed drafts of the article, and approved the final draft.
- Matthew M. Skinner conceived and designed the experiments, performed the experiments, analyzed the data, prepared figures and/or tables, authored or reviewed drafts of the article, and approved the final draft.

### Data Availability

The ply files for all images are available at The Human Fossil Record: https://human-fossil-record.org/index.php?/category/13466.

The specimen numbers and deposition information of the relevant specimens are available in the Supplemental File.

### Supplemental Information

Supplemental information for this article can be found online at http://dx.doi.org/10.7717/peerj.14523#supplemental-information.

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
