# Peer review of "Primate tooth crown nomenclature revisited"

_PeerJ, doi:10.7717/peerj.14523_

## Round 0.1 · original submission · Major Revisions

I apologize for the delay in my response, but I had some difficulty tracking down appropriate and willing reviewers over the summer. Two reviewers did respond and they both agree the manuscript has merit. While Reviewer 2 indicates only minor revisions, I suggest major revisions in to address the detailed concerns of Reviewer 1 in the review and the attached pdf. For example, please expand upon the justification or revise the taxonomic approach for the manuscript. I agree that an organization based on evolutionary clades (i.e., Strepsirrhini >... ...> Hominoids) as suggested by R1 as being more intuitive. However, if the order the results is based on another rationale (i.e. cusp simplicity > cusp complexity) this is fine but should be better explained in the Materials and Methods or beginning of the Results.

I also agree with Reviewer 2 that further justification is needed for why the study focuses only on the mandibular M1 and M2.

Both reviewers also provide a number of more detailed edits, suggestions, and concerns. If you feel these can be addressed, I look forward to receiving your revised manuscript.

Reviewer 1 ·

Excellent Review

This review has been rated excellent by staff (in the top 15% of reviews)
EDITOR COMMENT
This is a particularly insightful review that commented on both larger organizational concerns of the analysis as well as specific details that required deep knowledge of the field. I thank the reviewer for the their time and effort. Editors are dependent upon strong reviews like this one.

Basic reporting

As for the structure of the manuscript, I am wondering if there is any reason for the arrangement of the higher taxa (Cercopithecoids (Catarrhines) > Plathyrrhines > Prosimians > Hominoids (Catarrhines)) in the text and figures. Why were the two catarrhine groups separated widely? Perhaps a more ordinary arrangement is: Prosimii > Platyrrhines > Cercopithecoids > Hominoids or vice versa. Tarsiers are combined with lemurs, lorises and galagos in Prosimii in the manuscript, but they are phylogenetically closer to anthropoids than the latter. Considering that accessory cusp homology is a key point in this study, it seems more appropriate to use a taxonomy (Haplorhini and Strepsirrhini) that reflects more accurate phylogenetic relationships.

The taxomomy in Table 1 needs to be corrected. For example, Cercocebus is a papionin, not a cercopithecin. For details, see my comments in the pdf file. Also, check the taxonomy in your supporting file (accession numbers of all specimens studied), according to the comments in Table 1.

Figure 1
In this figure, the mesoconid is located within the taloned basin near the entoconid citing Van Valen (1966), but the mesoconid is located on the crista obliqua in Van Valen (1966)’s Fig.1, isn’t it?

There is a second entostylid located on the buccal cingulum citing Osborn (1888). Are you sure of the name and reference of this stylid?

Entostylid: Is Osborn (1888) the proper reference for this stylid? Not Osborn (1907), for example?

(Post)metaconulid: Why is “Post” in ( )?


Figure 3-19
The abbreviations such as BC, MM, BM, LM, DM in B) should be BCAC, MMAC, BMAC, LMAC, DMAC as they are in the 3D reconstructions of EDJ in the same figures.

Figure 7
As for LC in B), see my comment above for Figure 2.

Figure 12 and Lepilemuridae in the main text.
Metastylid in Schwartz and Tattersall (1985) = Entoconid in Swindler (2002), isn't it?
Swindler (2002) writes that lower M1-M2 have four cusps and M3 has a fifth cusp, the hypoconulid in Lepilemuridae. You are talking about M1-M2, not M3, aren’t you? I suppose neither Schwartz and Tattersall (1985) nor Swindler (2002) recognize the hypoconulid on M1-M2. So, the schematic drawings A and B are wrong in having hypoconulid (or distal margin accessory cusp).

Figure 14
Indri has bilophodont molars, in which there is a transverse crest between the hypoconid and entoconid apart from the distal marginal crest, as seen in this Figure 14D. So, the schematic drawings in A) and B) are OK for Propithecus and Avahi, but they are not adequate for Indri.

Figure 15B
Why is the crest between Hyd and End drawn straight, while it is slightly curved in Figure 15A?

Figure 16
Did previous researchers describe the lower molars of lorisids having the lingual cingulum? It is not mentioned in the main text.

Figure 18 and 19
What is the crest between the entoconid and hypoconid? In hominoids, an oblique crest connects the entoconid to the hypoconulid, forming a distal fovea, not to the hypoconid.

References
Sometimes bibliographical information is incomplete. I put some comments in the pdf file, but please check the references thoroughly.

Experimental design

This manuscript provides interesting information on the detailed morphology of the dentin enamel junction in major primate higher taxa, using CT-scans, with emphasis on accessary cusps. It is, however, regrettable that the present manuscript lacks the information about the outer enamel surface. Especially so, as the authors try to propose a new nomenclature system for accessory cusps. In most cases of research practices, it is difficult to observe the EDJ morphology, and what researchers can usually observe is the outer enamel surface morphology. Readers would be quite interested in how the EDJ variations revealed in the present manuscript are expressed in OES, especially those that the authors report as new variations. This manuscript would be more useful if information and discussion on OES and its relationships with EDJ are added.

Validity of the findings

I suppose that the presence of the lingual cingulum and the accessary cusps on it in the schematic in Figure 2 is based on a single specimen (ZMB 7888). It seems inappropriate to make a new scheme based on a single specimen, which could be aberrant. Or have you ever seen a certain number of other primate dental specimens through your macroscopic observation, which show such lingual cingulum and accessary cusps on it?

Annotated reviews are not available for download in order to protect the identity of reviewers who chose to remain anonymous.

·

Basic reporting

No comment

Experimental design

No comment

Validity of the findings

No comment

Additional comments

There are areas that need some clarifications and addition of a section. The authors have selected both first and second mandibular molars. However, other scholars have done studies related to cusp features both in the mandibular and maxillary molars (e.g., Martin et al., 2017; Ortiz et al., 2017). Thus, it would be useful to provide an explanation for selecting only the first and second mandibular molars in this study. Both the current and new variation schemes are clearly presented in the results and illustrated in the figures for each clade. This is great, but I recommend adding a table that summarizes both schemes.

Here are also some minor points.
1. Line # 98, author’s name and year should match with what is listed in the reference.
2. Line # 226, comma is needed after “ As a result”
3. Line # 227, “propose the follow terms” should be “propose the following terms”.
4. Line # 372, Cebinae should be replaced with Pitheciinae
5. Line # 446, cited author is not listed in the reference section.
6. Line # 472, “in this taxa” should be “ in this taxon”
7. Line # 758, year of publication is different from what is written in the in-text citation
8. Line # 781, there is no match between reference and the in-text citation
9. Line # 811, author is not included in the in-text citation
10. Line # 852, author is not included in the in-text citation

---

## Round 0.2 · Minor Revisions

Thank you very much for your revised manuscript. I sent it back to Reviewer 1 who had suggested Major Revisions for the initial submission. They now suggest minor revisions and I agree. There remains one unanswered question from their initial review. In addition, they have provided comments on the attached file with a number of smaller edits for spelling, grammar and reference formatting. Please address the remaining concern and minor edits. In addition, please thoroughly review your manuscript prior to resubmission. Once accepted, manuscripts move directly to the production stage, and it is my understanding there will only be one opportunity to make corrections at the page proof stage.

Reviewer 1 ·

Basic reporting

My previous comment
>There is a second entostylid located on the buccal cingulum citing Osborn (1888). Are you sure of the name and reference of this stylid? Entostylid: Is Osborn (1888) the proper reference for this stylid? Not Osborn (1907), for example?

There are still two “entostylid” in Figure 1. Although the authors corrected the references, they did not answer my question about the correctness of the stylid name. I would like to know whether Osborn (1888 or 1907) really referred to two different accessory cusps with the same name. If he really did so, which page(s) in Osborn (1888 or 1907) are these two 'entostylid' indicated?

There are still some minor in spelling and term usage. Please see the attached file.

Experimental design

The authors replied to my previous comment by providing 3D-reconstructed OES images in a supplementary file. For the purpose of this manuscript, I think it a satisfactory response.

Validity of the findings

The authors' reply to my previous comment about 'LCAC' in Cebinae is OK.

Annotated reviews are not available for download in order to protect the identity of reviewers who chose to remain anonymous.

---

## Round 0.3 · accepted · Accept

Thank you very much for your revised manuscript, it is ready to be accepted. Thank you very much for responding to the careful comments from the reviewers. Congratulations on your very nice analysis.